# Mechanisms of Global Ocean Ventilation Age Change during the Last Deglaciation

Lingwei Li[1], Zhengyu Liu[1], Jinbo Du[2], Lingfeng Wan[3,4], Jiuyou Lu[4]

[1]Department of Geography, The Ohio State University, Columbus, Ohio, 43210, United States
[2]Department of Atmospheric and oceanic sciences, Peking University, Peking, China
[3]Frontier Science Center for Deep Ocean Multispheres and Earth System (DOMES), Institute for Advanced Ocean Study (IAOS) and Key Laboratory of Physical Oceanography.MOE.China (POL), Ocean University of China, Qingdao 266100, China
[4]Laoshan Laboratory, Qingdao 266237, China

*Correspondence to*: Lingwei Li (li.8955@osu.edu)

**Abstract.** Marine radiocarbon ($^{14}$C) is widely used to trace deep ocean circulation, providing insight into the atmosphere-ocean exchange of $CO_2$ during the last deglaciation. Evidence shows a significant depleted $\Delta^{14}$C in the glacial deep ocean, suggesting an increased ventilation age at the Last Glacial Maximum (LGM). In this study, using two transient simulations with tracers of $^{14}$C and ideal age, we found that the oldest ventilation age is not observed at the LGM. In contrast, the models
show a modestly younger ventilation age during the LGM compared to present day. The global mean ventilation ages averaged below 1 km are approximately 800 (630) years and 930 (2000) years at the LGM and present day, respectively, in two simulations. This younger glacial ventilation age is mainly caused by the stronger glacial Antarctic Bottom Water (AABW) transport associated with sea ice expansion. Notably, the ocean ventilation age is significantly older predominantly in the deep Pacific during deglaciation compared to the age at the LGM, with global mean ventilation ages peaking at 1900
and 2200 years around 14-12 ka in two simulations, primarily due to the weakening of AABW transport.

## 1 Introduction

Discussion about past and future climate change is often difficult without reference to the global ocean circulation, because of its critical role in storing and transporting heat, carbon, and nutrients. Ice core records demonstrate significant variability in atmospheric carbon dioxide ($CO_2$) concentrations on the glacial-interglacial timescale (Sigman and Boyle, 2000; Monnin
et al., 2001; Schmitt et al., 2012). During the last glacial period, atmospheric $CO_2$ levels were approximately 90 parts per million by volume (ppmv) lower than the Holocene value of 280 ppmv (11.7-0 ka where ka indicates "thousand years ago). This variation in $CO_2$ is closely coupled with long-term climate changes and the carbon cycle in the Earth system. Various oceanic processes thus have been suggested to regulate atmospheric $CO_2$, such as air-sea gas exchange (Long et al., 2021), biological production (Broecker, 1982; Sigman et al., 2021; Sigman and Boyle, 2000), and ocean circulation (Ai et al., 2020;
Marcott et al., 2014; Tschumi et al., 2011; Skinner et al., 2015, 2010). Consequently, understanding past changes in global

ocean circulation provides a good opportunity, not only for a more accurate constraint on the past climate change and carbon cycle, but also for insight into future climate change with rising atmospheric $CO_2$ level.

Various proxy reconstructions have been used to investigate past changes in oceanic circulation. Notably, radiocarbon ($\Delta^{14}C$) observations are valuable for determining changes in the deep ocean circulation in terms of ventilation age, which refers to the "age" of seawater as the time elapsed since the last contact of a water parcel with the atmosphere. Evidence indicates that, during the Last Glacial Maximum (LGM; 23-18 ka), the North Atlantic and North Pacific exhibited significant depletion in $\Delta^{14}C$ below approximately 2.5 km compared to present day levels (Marchitto et al., 2007; Rafter et al., 2022; Skinner et al., 2015, 2017). A comparison with other radiocarbon data from the southern high-latitudes suggests consistent changes experienced across the Southern Ocean (Burke and Robinson, 2012; Chen et al., 2015; Schmitt et al., 2012; Skinner et al., 2010). This $\Delta^{14}C$ depletion has been suggested as an indication that the deep waters were isolated and poorly ventilated, leading to an increased ventilation age at the LGM relative to the present day.

The relationship between the $\Delta^{14}C$ depletion at the LGM and the true ocean ventilation time, however, has remained uncertain. Traditionally, two ages can be estimated from the ocean $\Delta^{14}C$ at depth: the $\Delta^{14}C$ B-A age that is estimated from the benthic-atmosphere $\Delta^{14}C$ age difference, and the $\Delta^{14}C$ B-P age that is estimated from the benthic-planktonic $\Delta^{14}C$ age difference, with the difference between the two ages caused by the surface water, or the marine reservoir age. The temporal evolution of the atmospheric $\Delta^{14}C$ during the deglaciation further leads to a changing source of ocean water $\Delta^{14}C$, which leads to the so called projection age (Adkins and Boyle, 1997). An analysis of $\Delta^{14}C$ B-P age in the deep Pacific suggests that ventilation age increased by ~1000 years during the deglaciation, contradicting the expected pattern of an isolated carbon reservoir in the glacial deep North Pacific (Lund et al., 2011). Climate models with radiocarbon have also been used to study ocean ventilation age during the deglaciation. Model simulation shows that sea ice expansion at the LGM contributes to a depletion of ocean $\Delta^{14}C$ and in turn an older radiocarbon age than the true ventilation age (Schmittner, 2003). Recently, Zanowski et al. (2022) assessed the relationship between the evolution of $\Delta^{14}C$ and circulation changes in the deep Pacific Ocean in a transient ocean simulation forced by realistic climate forcing during the last deglaciation (22–0 ka): the C-iTRACE ocean model simulation (Gu et al., 2020, 2021b, 2019a). They found that the deglacial variations in Pacific Ocean $\Delta^{14}C$ age are influenced by changes in both the deep circulation and surface ocean reservoir age. Therefore, the $\Delta^{14}C$ age could differ substantially from the true ocean ventilation age.

In this study, we examine the global ocean ventilation age in the C-iTRACE simulation. Figure 1a shows the deglacial evolutions of the B-A age and B-P age averaged globally below 1 km. The B-A age is calculated as $8267 \times ln(\frac{\frac{\Delta^{14}C_{atmosphere}}{1000}+1}{\frac{\Delta^{14}C_{ocean}}{1000}+1})$, where $\Delta^{14}C_{atmosphere}$ is the $\Delta^{14}C$ in the atmosphere, and the B-P age is calculated as $8267 \times ln(\frac{\frac{\Delta^{14}C_{surface}}{1000}+1}{\frac{\Delta^{14}C_{ocean}}{1000}+1})$, where $\Delta^{14}C_{surface}$ is the $\Delta^{14}C$ average value of upper 100 m of the ocean. These two ages are compared with the true global ocean ventilation age in the model, or the ideal age (IAGE). First, both the B-A age and B-P age are

considerably older at the LGM (~20 ka) than the preindustrial period (denoted here as the present day or PD, ~0 ka), and show decreasing trend from the LGM towards the PD. This is largely consistent with the observations of more depleted deep water $\Delta^{14}$C at the LGM (Burke and Robinson, 2012; Chen et al., 2015; Okazaki et al., 2010; Rafter et al., 2022; Skinner et al., 2017, 2019). Second, both the B-A age and B-P age are significantly older than the IAGE, with dramatically different deglacial evolution pattern. The IAGE is slightly younger at the LGM than at the present and ages dramatically by over two times towards before the Younger Dryas (YD; 12.9-11.7 ka), in sharp contrast to the radiocarbon ventilation ages that showed minimal change before the YD and then a predominately decreasing trend in the Holocene. Similar behaviours in radiocarbon ages and IAGE are also present in the Pacific and Atlantic Oceans (Fig. 1b-c). These radiocarbon ages raise two questions in the model context. First, why are the radiocarbon ages different from the true ventilation age, and how to estimate the true ocean ventilation age from the ocean radiocarbon? Second, why is the true ocean ventilation age comparable between the LGM and the present, and why does it increase to its oldest age before the YD?

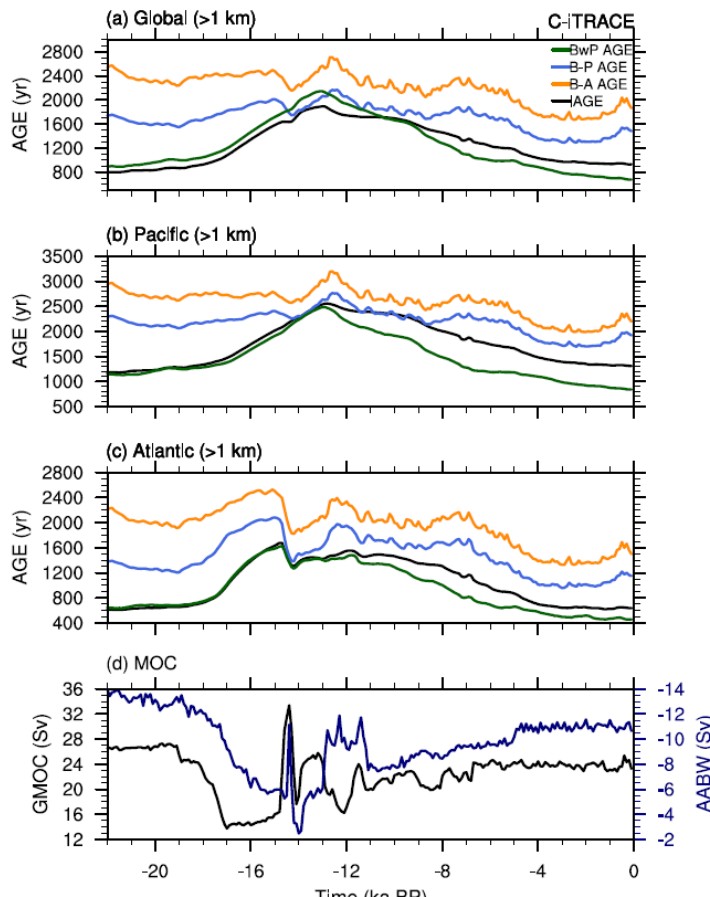

**Figure 1 Time evolutions in the C-iTRACE: (a) The global mean ideal age (IAGE; black), benthic-atmosphere $\Delta^{14}$C age (B-A age; yellow), benthic-planktonic $\Delta^{14}$C age (B-P age; blue), and weighted benthic-planktonic $\Delta^{14}$C age (BwP age; green) averaged below 1 km. (b) The Pacific mean IAGE (black), B-A age (yellow), B-P age (blue), and BwP age (green) averaged below 1 km. (c) The**

**Atlantic mean IAGE (black), B-A age (yellow), B-P age (blue), and BwP age (green) averaged below 1 km. (d) The Global Meridional Overturning Streamfunction (GMOC; black) and Antarctic Bottom Water strength (AABW; navy). Here the GMOC intensity is diagnosed as the maximum in the GMOC streamfunction below 600 m from 33° S-60° N, and AABW is diagnosed as the minimum in the GMOC streamfunction below 2 km over 2° S-70° S.**

The first question will be discussed in a separate paper (Du et al., under review), which proposes a new method to estimate the ventilation age from radiocarbon using the so called weighted benthic-planktonic $\Delta^{14}$C age (BwP age) by taking account of multiple water mass contributions (Fig. 1a-c green lines). This new approach of BwP age is similar to the ideal age (IAGE) globally, suggesting that water age is not old at the LGM and, instead, peaks in the middle of deglaciation around 13 ka. The similar temporal evolution patterns of BwP age and model ventilation age suggest that B-P and B-A ages are likely biased by remote water mass sources (e.g. Antarctic Bottom Water from the Southern Ocean) and can be greatly improved. Due to the potential challenges in radiocarbon proxies to accurately estimate the deep ocean ventilation age, we will address the second question regarding the mechanism of the deglacial evolution of the model ventilation time, in order to provide additional model perspectives on changes in the global ocean ventilation age during the last deglaciation.

In this study, we show that the deep ocean ventilation time in the model is determined mainly by the circulation of the Antarctic Bottom Water (AABW) with the deep Pacific playing the dominant role. We will describe the model simulation and method in section 2. The mechanism of the evolution of the ocean ventilation time is discussed in detail for C-iTRACE in section 3 and briefly for another simulation iTRACE in section 4. A summary is given in section 4.

## 2 Models and data

We will analyze the C-iTRACE simulation, which is an ocean-only deglacial transient simulation (22-0 ka) in the isotope-enabled version of the Parallel Ocean Program version 2 (POP2; Danabasoglu et al., 2012), the ocean component of the Community Earth System Model (CESM). The model configuration comprises 60 vertical layers and a nominal 3° horizontal resolution. The C-iTRACE is forced by the monthly surface forcings (heat flux, freshwater flux, and momentum flux) from a fully coupled transient simulation (TRACE-21ka), which is forced by realistic external forcing of continental ice sheet, greenhouse gases, orbital forcing, and melting water fluxes and simulates many key features of the last deglaciation (Liu et al., 2009). The simulation incorporates multiple geotracers, such as carbon isotopes (Jahn et al., 2015), neodymium isotopes (Gu et al., 2019b), $^{231}$Pa/$^{230}$Th ratio (Gu and Liu, 2017), and oxygen isotopes (Zhang et al., 2017). The incorporation of geotracers in the C-iTRACE facilitates a direct model-data comparison, and the transient simulation C-iTRACE has been used in many recent studies to understand intermediate and deep water masses and oceanic circulation variations (Gu et al., 2020, 2021b; Zanowski et al., 2022; Zhang et al., 2017). As the C-iTRACE simulation has been explored in greater detail in recent studies, including the comparison of the simulated radiocarbon ages and proxy observations, this study is a pure modelling study aiming to understand ventilation ages in the deep ocean over the last deglaciation.

To further improve the understanding of deglacial deep ocean ventilation changes and associated mechanisms, several idealized tracers are also implemented in C-iTRACE. First, the ideal age (IAGE) is included, which is set to 0 at the ocean

surface and ages at a rate of 1 year/year thereafter passively advected and diffused into the ocean interior. Thus, IAGE is a passive circulation tracer measuring the time elapsed since the last contact with the atmosphere and working like a clock counting time after being restored to zero (England, 1995; Koeve et al., 2015). In turn, the IAGE represents the "true" model ventilation age, although it does not account for the insulation effect of sea ice. Second, several dye tracers are implemented in the model to identify water mass composition. The dye tracers are reset to 1 over specific region at the ocean surface during each time step and passively advected and diffused within the ocean interior. In this study, two dye tracers are used over the surface Southern Ocean (south of 34°S, Dye_S) and the North Atlantic (north of 40°N, Dye_NA). Detailed descriptions of these tracer experiments can be found in previous manuscripts (Gu et al., 2020, 2021b; Zanowski et al., 2022).

In addition to C-iTRACE, we also analyze the deglacial ventilation age changes in another fully-coupled deglacial simulation, iTRACE, which is conducted with the isotope-enabled CESM (iCESM, Brady et al., 2019). The iTRACE experiments are performed following a similar strategy as in the previous transient simulation TRACE-21ka (Liu et al., 2009), but in the state-of-the-science CESM1.3 (Hurrell et al., 2013). Similar to C-iTRACE, the ocean component of iTRACE consists of 60 vertical layers, but with a higher nominal 1° horizontal resolution. It also includes multiple geotracers, such as radiocarbon, oxygen isotopes, and IAGE, although, unfortunately, the radiocarbon output is only available from 20 ka to 11 ka in this paper. The iTRACE has been directly compared to multiple proxy reconstructions and is able to quantitatively capture many major features of climate variations during the last deglaciation (Brady et al., 2019; Hurrell et al., 2013). Further details regarding the iTRACE simulation can be found in previous manuscripts (He et al., 2021a, b).

In the following analyses, the time resolution for all variables is 100-year average in both simulations, and all times (calendar ages) are reported in thousands of years before present (ka BP where BP indicates 1950 CE). Climatologies for the LGM and PD states in the C-iTRACE are taken as the 100 year means of 20 ka and 0 ka respectively.

## 3 C-iTRACE results

### 3.1 LGM and PD ventilation ages

In the C-iTRACE simulation, a modestly younger ventilation age (IAGE) can be observed in the deep ocean at the LGM compared to the PD. Specifically, at depths below 1 km, the global mean IAGE is 800 years at the LGM, whereas it is 930 years at the PD (Fig. 1a). Similarly, in the Pacific region, the IAGE is 1181 years at the LGM, which is younger than the age of 1310 years observed during the PD (Fig. 1b). This modestly younger LGM IAGE is also evident in the comparison of global zonal mean distributions of IAGE between the LGM and PD (Fig. 2a and 2c). The maximum IAGE in the deep ocean is 1660 years during the LGM and 1763 years during the PD. Therefore, it suggests that the deep ocean is well ventilated at the LGM than today in the model.

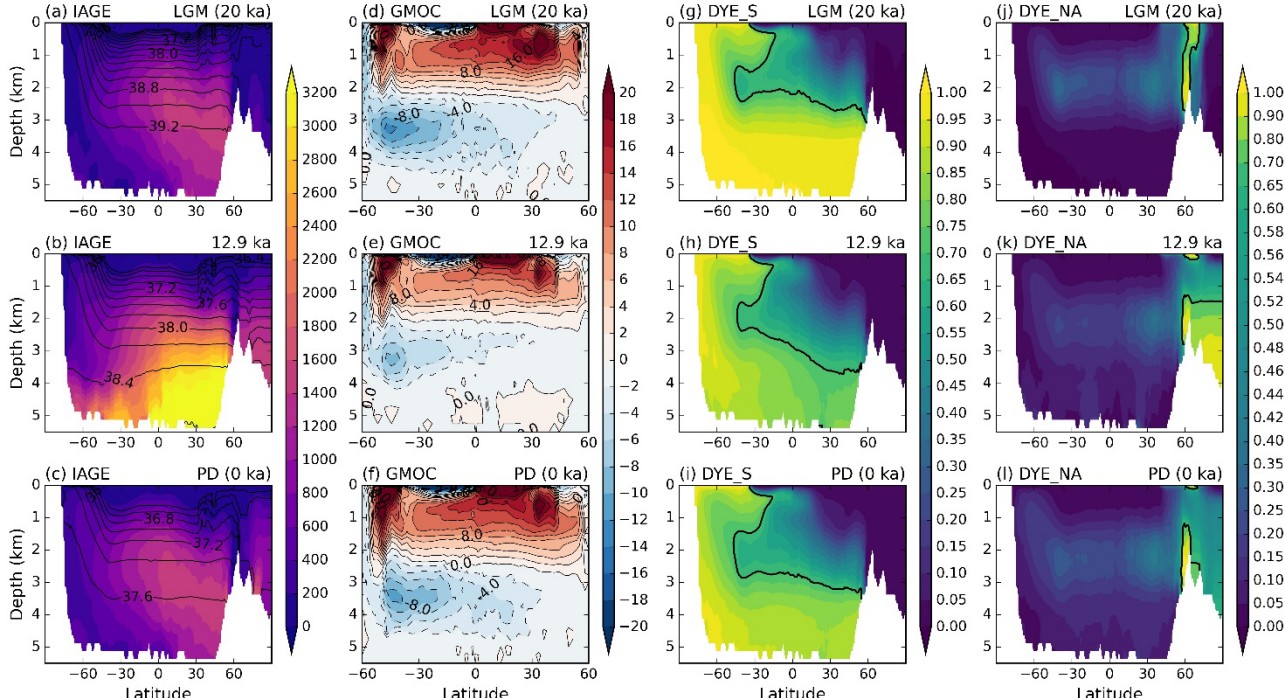

**Figure 2 The global zonal mean distributions at LGM (20 ka), 12.9 ka, and present day (PD, 0 ka). (a-c) IAGE (shading; unit in year) with isopycnals (σ2; potential density referenced to 2000 m; black contour lines; unit in kg/m³). (d-f) Global residual meridional overturning circulation streamfunction (GMOC; unit in Sv). (g-i) Dye-South : percentage of water originating from the Southern Ocean. (j-l) Dye-North: percentage of water originating from the North Atlantic. The thick black contours on dye tracers (g-l) represent the value of 0.7. Note the nonlinear magnitude of the colour bar for Dye-North.**

To elucidate the reasons behind the younger LGM IAGE, we first explore whether it is likely attributed to the AABW transport. At present day, the deep ocean circulation generally consists of two cells (Cessi, 2019; Marshall and Speer, 2012; Talley, 2013). The upper cell, confined to the Atlantic basin, begins with the formation of deep water North Atlantic Deep Water (NADW). The NADW sinks into deeper depth at the northern Atlantic and flows southward at middepth in the basin primarily along deep western boundary all the way to the Southern Ocean, and eventually upwells along isopycnals and returns back northward as intermediate waters. The abyssal cell, most prominent in the Indo-Pacific sector, starts with the bottom water AABW formation in the Southern Ocean, which sinks from its formation sites around the Antarctica and moves northward across topography to fill the deepest parts of the global ocean before upwelling to middepth and returning to the surface along isopycnals in the Southern Ocean. Given the substantial volume of the AABW and its role in driving the abyssal cell, it is reasonable to hypothesize that a reduction in AABW volume transport is linked to a decrease in abyssal ocean ventilation rate and, in turn, an increase of ventilation time.

This hypothesis is supported by the C-iTRACE analysis. In the model, the maximum magnitude in the abyssal global residual Meridional Overturning Circulation (GMOC) is 13.1 Sv (1 Sv ≡ $10^6$ m³ s⁻¹) at the LGM and 10.6 Sv at the PD (Fig. 2d and 2f). The intensified LGM AABW transport in the C-iTRACE is consistent with reconstructions and modelling studies

showing larger salinification of AABW (Adkins et al., 2002; Hesse et al., 2011; Negre et al., 2010; Schmittner, 2003). Model simulations further suggest that the strong glacial AABW transport is dominated by changes in surface buoyancy forcing over the Southern Ocean (Ferrari et al., 2014; Jansen, 2017; Jansen and Nadeau, 2016; Liu, 2023; Shin et al., 2003; Sun et al., 2018). The sea ice expansion at the LGM enhanced the brine rejection during winter, leading to extremely saline and dense AABW. Thus, the glacial deep ocean is filled with greatly expanded cold-salty AABW water mass with higher densities, contributing to a stronger AABW transport. The stronger glacial AABW transport, in turn, reflects a shorter residence time in the ocean interior, as indicated by the overall younger LGM IAGE compared to the water age at the PD.

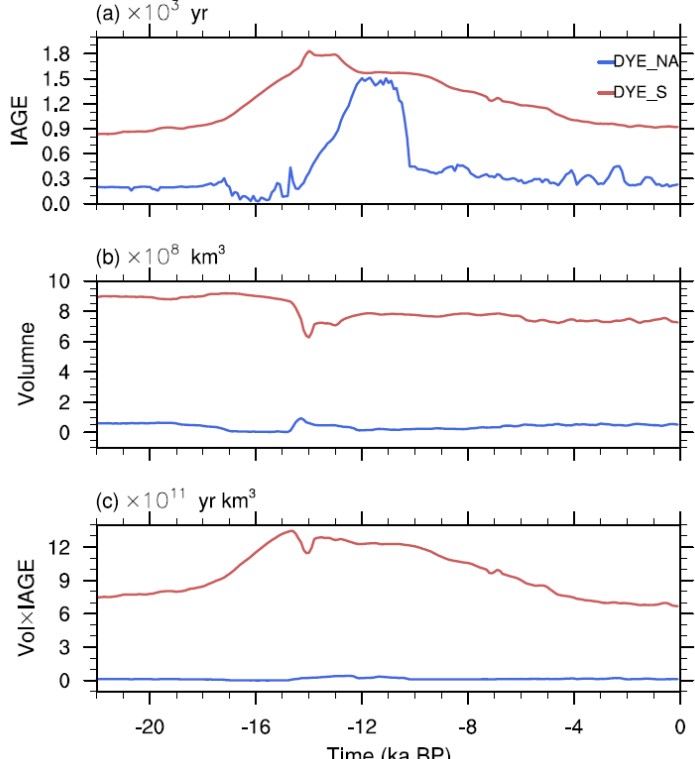

**Figure 3 Time evolutions for water masses sourced from different regions. (a) The global volume-weighted average of ideal age (IAGE) where fractions of water originating from the Southern Ocean (red) and North Atlantic (blue) are greater than 70%. (b) The total water volume where fractions of water originating from the Southern Ocean (red) and North Atlantic (blue) are greater than 70%. (c) The age of total volume of water from the Southern Ocean and North Atlantic (the product of volume from (b) and the global averaged ideal age from (a)).**

The dominant role of the AABW transport in determining the ventilation age at the LGM is also evident in the idealized dye concentrations. Since the water mass distributions can be identified unambiguously using dye tracers, the comparison between the IAGE and dye tracers shows the younger LGM IAGE is likely associated with vigorous AABW. Below 3 km, the average proportion of AABW (AABW%) represented by dye concentration from the Southern Ocean (Dye_S) is 86% at the LGM (Fig. 2g), which is 16% higher than that observed at the PD (Fig. 2i). This high AABW% coincides with the strong northward AABW flow (Fig. 2d) and the notable young ventilation age in the deep ocean at the LGM (Fig. 2a). In contrast,

the fraction of NADW (NADW%), indicated by dye concentration from the North Atlantic (Dye_NA), remains relatively low at 2% and 18% below 3 km during the LGM and PD, respectively, except in the Arctic Ocean (Fig. 2j and 2l). Thus, the fractions of NADW and AABW illustrate the dominant role of AABW, in comparison to NADW, in filling the abyssal ocean and influencing deep ocean ventilation age.

    The dominant role of AABW in deep ventilation age at the LGM can be further quantified by calculating the deep ventilation 185     age of the AABW and NADW water masses. The IAGE value is calculated where the percentages of AABW% and NADW% exceed 70% below 1 km in the global ocean (Fig. 3a). The conclusion is somewhat insensitive to the choice of dye concentration value for NADW and AABW. A higher (lower) value of 80% (60%) indicates less (more) mixing with other water masses and thus does not affect the main results. The averaged IAGE for AABW is about 833 years at the LGM and 921 years at the PD, while the IAGE for NADW is about 200 years at both the LGM and PD. Furthermore, the volume of 190     AABW is about $9 \times 10^8$ km$^3$ at the LGM and PD, such that the volume integrated IAGE for AABW is about $7 \times 10^{11}$ yr km$^3$ at both LGM and PD (Fig. 3b and 3c). Thus, the younger ventilation age in the deep ocean at the LGM is caused by a relatively younger AABW at the LGM than the PD.

## 3.2 Mechanisms for deglacial evolution of ventilation age

    In comparison with the almost comparable IAGEs of LGM and PD, the deglacial evolution exhibits a bell shape, with the 195     global deep IAGE increasing from 800 years at the LGM to its oldest value of more than doubled to 1900 years at 12.9 ka, followed by a decline back towards 930 years in the Holocene period (Fig. 1a). A similar bell shape pattern is also seen for the Pacific mean (Fig. 1b). This dramatically older deep water at 12.9 ka is also seen clearly in comparison of the oldest water in the global zonal mean IAGE, which is 3847 years at 12.9 ka, more than doubling the oldest ages of 1660 years at the LGM and 1763 years at the PD (Fig. 2a-c). This deglacial evolution of global IAGE can be also seen in the depth-time 200     Hovmöller diagram of global IAGE anomaly relative to the PD value (Fig. 4a). Across the depth, the ventilation age is relatively younger at the LGM in the deep ocean, mainly because of strong AABW as discussed in section 3.1. More importantly, the IAGE tends to increase monotonically from the LGM to Heinrich Stadial 1 (HS1; 17.5-14.7 ka), reaching to its oldest value in the deep ocean around 12.9 ka, and subsequently becoming younger toward the PD. This bell shaped deglacial evolution of global IAGE aligns with similar transport evolutions of the residual GMOC and global AABW (Fig. 205     1d, Fig. 2d-f) as well as the AABW% in dye tracer from the Southern Ocean (Fig. 3, Fig. 2g-i). Here, the GMOC is diagnosed as the maximum in the GMOC streamfunction below 600 m from 33° S-60° N and, therefore, the main feature of the GMOC follows the upper clockwise cell mostly confined to the Atlantic sector. Since the abyssal ocean is predominately filled by AABW, the evolution of IAGE corresponds more closely to the AABW intensity than GMOC. As shown in figure 1, starting from 19 ka, the GMOC and AABW both fall rapidly and then shows a sharp recovery. When the AABW transport 210     reaches its minimum strength at approximately 14 ka, the global mean IAGE increases towards its maximum at 12.9 ka. The lag between the AABW transport minimum at 14 ka and IAGE peak at 12.9 ka is likely because the "memory" of ocean typically last thousands of years. That is, the response time scale for the slow evolution of circulation associated with the

AABW and abyssal flows can be over a thousand years. As such, the decrease in global AABW transport tends to align with the increase in global mean IAGE more closely during the same period. Therefore, this evolution appears still consistent with
215 the hypothesis that the deep ventilation time is controlled mainly by the AABW transport during the deglaciation.

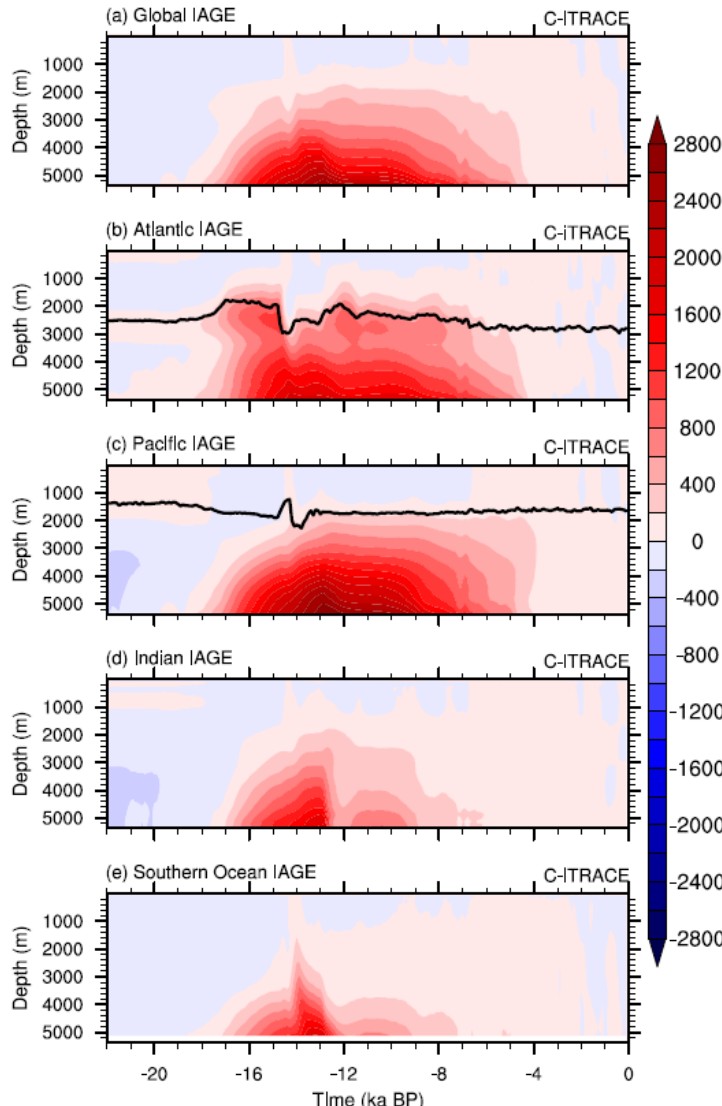

**Figure 4 Depth-time Hovmöller diagram of IAGE anomalies relative to present day over global and different basins. (a) The global mean. (b) Atlantic basin average. (c) Pacific basin average. (d) Indian basin average. (e) Southern Ocean average. The thick black lines on (b) and (c) are the depth of the zero-contour of Atlantic and Indo-Pacific Meridional Overturning Streamfunction**
**averaged between 60° N and 30° S respectively.**

To further understand how the AABW variation changes deep ventilation time during the deglaciation, the ventilation age and AABW transport across different ocean basins are analyzed. The time evolutions of IAGE in each basins show similar bell-shaped patterns, with the ventilation age increasing in the deep ocean during deglaciation, towards its maximum at 14.5

ka in the Atlantic and 12.9 ka in the Pacific, and then decreasing towards the Holocene (Fig. 4b-e). Relatively, the oldest

water always occurs in the deep Pacific, about 700 years older than the deep Atlantic with the largest volume, suggesting that

the Pacific Ocean dominates global ocean ventilation age, harbouring the oldest water mass mixture during the last

deglaciation.

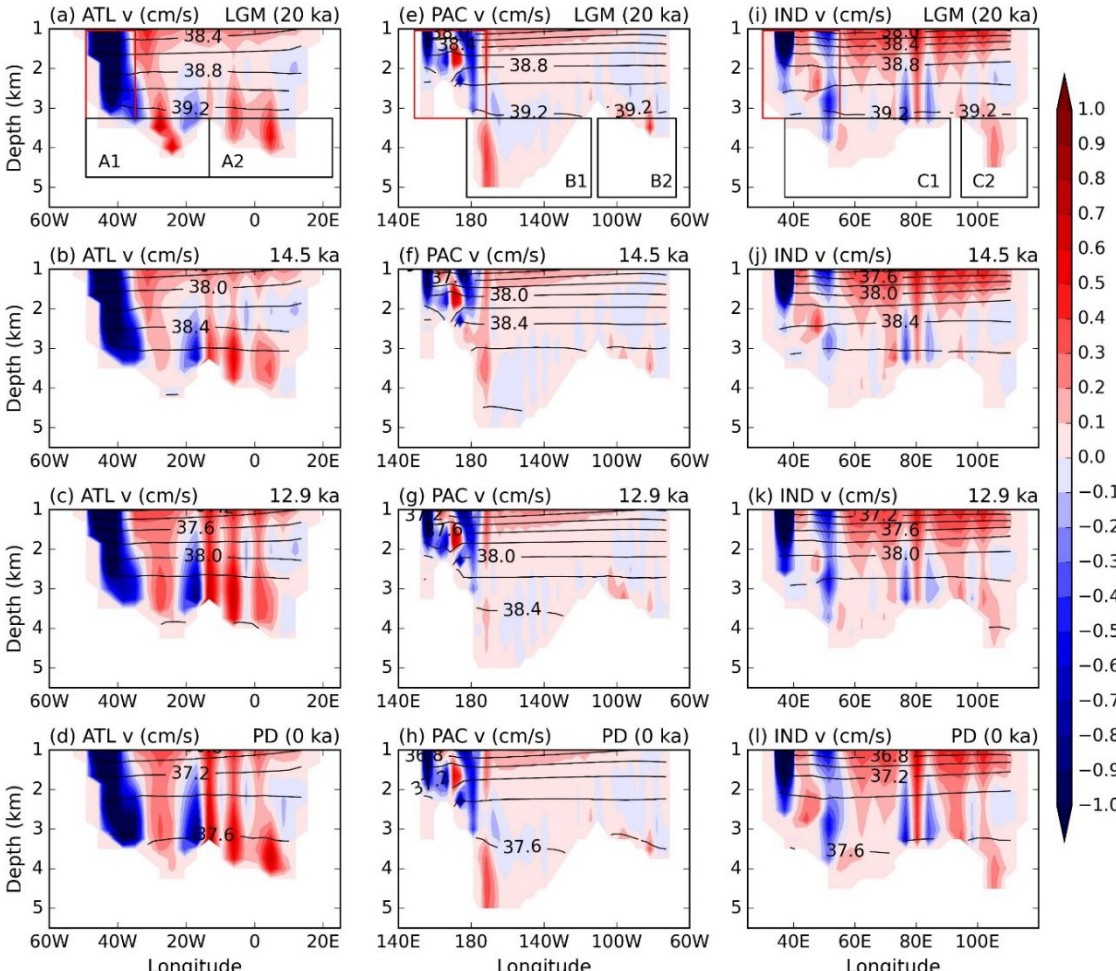

**Figure 5 Meridional velocity (shading) and potential density (lines) referenced to 2000 m at 30° S during the LGM (20 ka), 14.5 ka,**
**12.9 ka, and PD (0 ka) in the C-iTRACE. The shading contour interval is 0.1 cm/s and the black line contour interval is 0.2 kg/m³.**
**The southward deep western boundary current is defined as the southward integrated volume transport within the highlighted red**
**box on (a, e, i) for the Atlantic, Pacific, and Indian Oceans respectively. The export pathways of AABW in each basin is calculated**
**as the northward integrated volume transport within the black boxes on (a, e, i), noted as A1 and A2 for the Atlantic, B1 and B2**
**for the Pacific, and C1 and C2 for Indian Ocean.**

More quantitatively, the AABW transport in each basin will be calculated individually to verify the hypothesis. Along the

western boundary of each basin, deep western boundary currents (DWBCs) of northward AABW are observed at different

times, notably at the LGM, 14.5 ka, 12.9 ka, and PD, although the currents are strongly diffused by the coarse model

resolution (Fig. 5). In the Atlantic below 1 km, the southward DWBC carries the NADW from the subpolar North Atlantic while the northward abyssal DWBC against the continental slope carries the AABW northward, with the AABW DWBC

lying beneath and offshore of the NADW DWBC (Fig. 5a-d). In the Pacific and Indian Oceans, only northward DWBCs are present in the abyssal layers, carrying the dense AABW northward (Fig. 5e-l). Moreover, the meridional velocity for bottom AABW flow weakens at 14.5 ka in the Atlantic basin (Fig. 5b), and at 12.9 ka in the Pacific and Indian Oceans (Fig. 5g and 5k), consistent with the IAGE trend shown in Figure 4b and 4c. This supports the bottom AABW current as the primary cause of the deglacial increase in ventilation age.

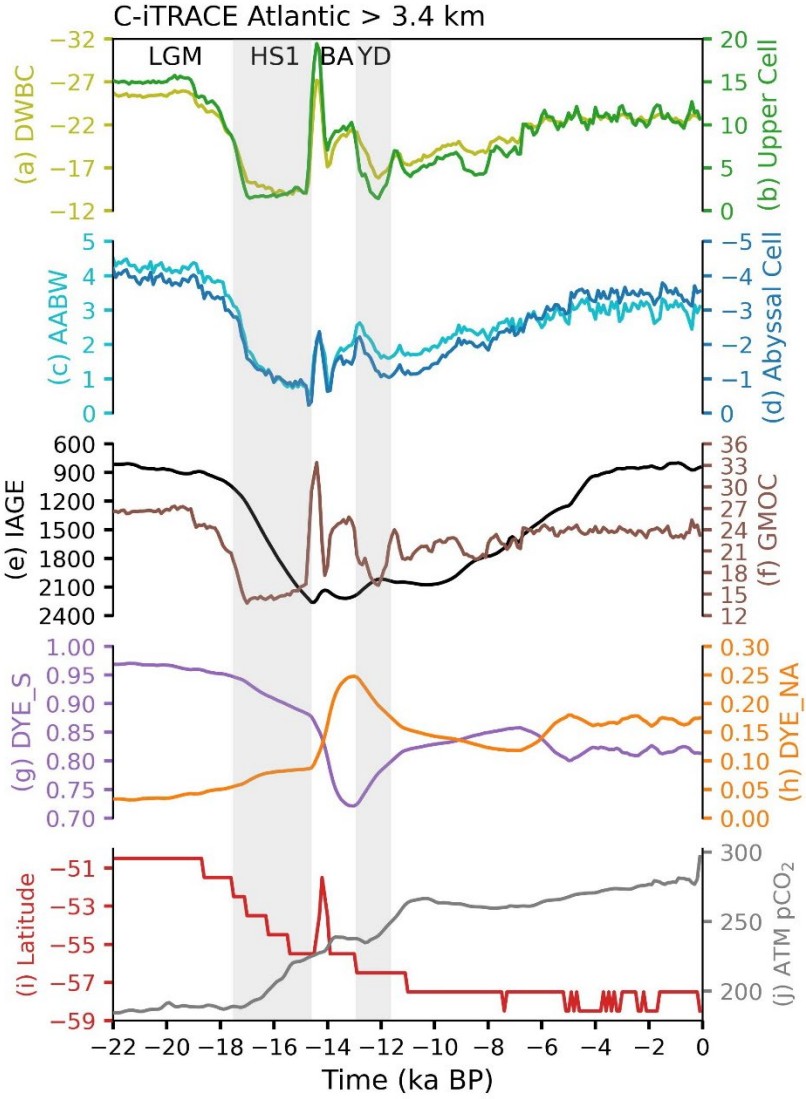

**Figure 6 Atlantic deglacial evolutions in the C-iTRACE: (a) Southward Deep Western Boundary Current (DWBC; light olive, unit in Sv) defined as the southward integrated volume transport within the red box in Figure 5, (b) Upper cell (green, unit in Sv) defined as the maximum in Atlantic Meridional Overturning Circulation (AMOC) between 0.6-3.5 km at 30° S, (c) AABW**

transport (cyan, unit in Sv) defined as $\Psi_{AABW} = -\int_{z=-H}^{z=-3.4km} \int v\,dxdz$ where v is meridional velocity component and H is the bottom seafloor, (d) abyssal cell (blue, unit in Sv) defined as the minimum in AMOC below 2.5 km m at 30° S, (e) IAGE (black, unit in yr) averaged below 3.4 km, (f) GMOC strength (brown, unit in Sv) defined the same as that in Figure 1d, (g) Dye-South (purple, unit in %) averaged below 3.4 km, (h) Dye-North (yellow, unit in %) averaged below 3.4 km, (i) the latitude of 20% sea ice coverage averaged over Southern Ocean (red), and (j) The atmospheric $CO_2$ (gray, unit in ppmv). The gray shadings delimit periods of abrupt change. LGM-Last Glacial Maximum; HS1-Heinrich Stadial 1; BA-Bølling-Allerød; YD-Younger Dryas.

As northward abyssal DWBC AABW is identified across section at the southern edge 30° S, the AABW transport in each basin is calculated from the meridional velocity at 30° S, using the basin wide integrated volume transport below 3.4 km as $\Psi_{AABW} = -\int_{z=-H}^{z=-3.4km} \int v\,dxdz$ where v is meridional velocity component and H is the depth of the seafloor. This is because the isopycnals (potential density referenced to 2000 m) across each basin at 30°S are mostly flat below 3.4 km, and the bottom AABW currents are strongly diffused by the coarse model resolution with very weak interior flow (Fig. 5). The southward DWBC in the deep to intermediate layers is thus defined as the southward integrated volume transport within the red box shown in figure 5a, 5e, and 5i. The estimations of southward DWBC and northward AABW DWBC are confirmed by the model MOCs (Upper and Abyssal cells in the figures). The upper cell is diagnosed as the maximum in the overturning streamfunction between 0.6–3.5 km at 30° S and the abyssal cell is diagnosed as the minimum in the overturning streamfunction below 2.5 km at 30° S. The calculated northward AABW DWBC transport aligns exceptionally well with the transport of model abyssal cell in each basin (Fig. 6a-b, Fig. 7a-b). The transports of the southward DWBC and model upper cell are differed by the northward return flow in the interior ocean (Fig. 6c-d, Fig. 7c-d).

In the Atlantic basin, the deglacial increase in ventilation age is tied to the reduction of AABW transport. The IAGE averaged across the Atlantic basin below 3.4 km increases from 820 years at the LGM to its oldest age of 2260 years at 14.5 ka, followed by a decrease back towards 840 years at the PD (Fig. 6e). The northward AABW DWBC transport tracks the evolution of IAGE more closely, as AABW transport slows to 0.2 Sv at 14.5 ka and then recovers to 3 Sv at the PD (Fig. 6c). Therefore, the weakening of AABW transport during deglaciation corresponds to a slower ventilation rate and in turn an older ventilation age in the abyssal Atlantic basin.

Transient dye tracer concentrations further support the dominant role of the AABW transport in determining deglacial deep ventilation age. The averaged AABW% declines from LGM to 12.9 ka, indicating a reduced AABW transport flowing into Atlantic (Fig. 6g). In the meantime, the NADW% rises to 25% from LGM to 12.9 ka, contributing to a relatively small portion of younger water flushing the deep Atlantic (Fig. 6h). Note that the ventilation age of AABW water mass is typically much older than the age of NADW (Fig. 3). This is because the NADW water mass characterizes the southward branch of the upper cell confined to the Atlantic sector, which is interconnected with AABW and thermocline ventilation, leading to relatively short residence times through the ocean interior. The mean IAGE for AABW water mass increases from 836 years at the LGM to 1813 years at 14 ka due to the weakening of AABW transport, while the age of NADW increases by up to 1500 years during the period of 12–11 ka. The increased water age of NADW is attributed to a greater fraction of NADW sinking into deeper Arctic depths (Fig. 2j-l) resulting in an increased water age for NADW as distance from the formation regions increases and the lag of 2000 years between the maxima of Dye_NA and Dye_S (Fig. 3a-b). However, because

AABW% in the abyssal southern Atlantic is substantially higher than NADW% (Fig. 6g-h), the presence of the oldest water
at 14.5 ka can be mainly attributed to the weakened AABW transport.

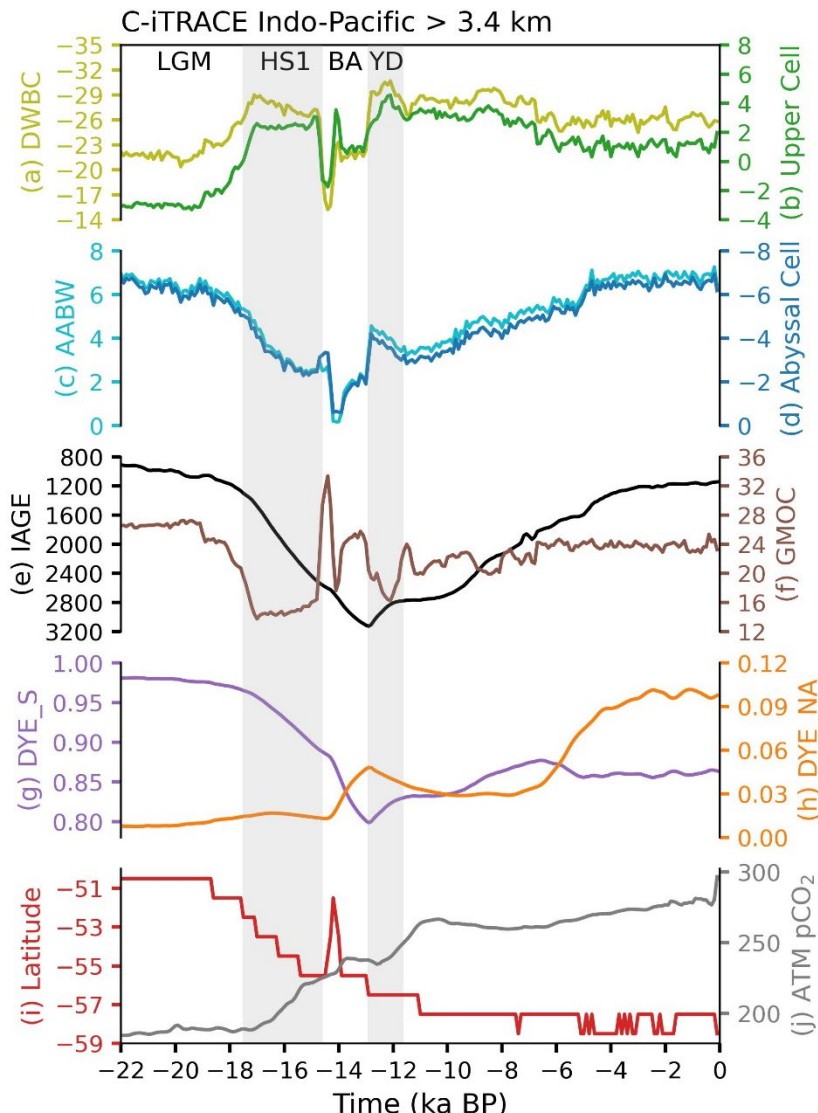

**Figure 7 Similar to Fig. 6 but for Indo-Pacific basin.**

The influence of weakened AABW on the deglacial increase in ventilation age is also prominently evident in the Indo-Pacific sector. The IAGE averaged across the Indo-Pacific basin becomes older from the LGM to 12.9 ka, coinciding with
the decrease in northward AABW DWBC transport to 2.1 Sv at 12.9 ka (Fig. 7e-f). In addition, different from NADW%, the AABW% decreases to its minimum value at 12.9 ka, indicating a reduced AABW transport flowing into the Indo-Pacific basin at that time (Fig. 7g-h). Therefore, the deglacial increase of IAGE in the Indo-Pacific can be mostly explained by the weakening of AABW.

In addition to the AABW transport variations in each basin, the export pathways of AABW from the Southern Ocean into each basin are also explored. Particularly, branches of AABW transport in each basin is calculated as the northward integrated volume transport at 30° S within the black boxes shown in Figure 5a, 5e, and 5i. The deglacial changes of each AABW branch are shown in Figure 8. In the Atlantic at 30° S, the main export route occurs within broad belts east of the Mid Atlantic Ridge (Fig. 8a). The Pacific AABW flows northward along the southwest Pacific basin and is a major source of waters ventilating the deep Pacific basins (Fig. 8b). In the Indian Ocean, however, the AABW transport is relatively smaller than that in the Atlantic and Pacific Oceans, and deep pathway of AABW is mainly along the eastern Indian Ocean basin (Fig. 8c). The AABW pathways in the Pacific are generally consistent with the available modern observations (Purkey et al., 2018), but different circulation routes of AABW in the Atlantic and Indian Oceans are found in the C-iTRACE, probably due to model deficiencies such as the coarse resolution and eddy-parameterization.

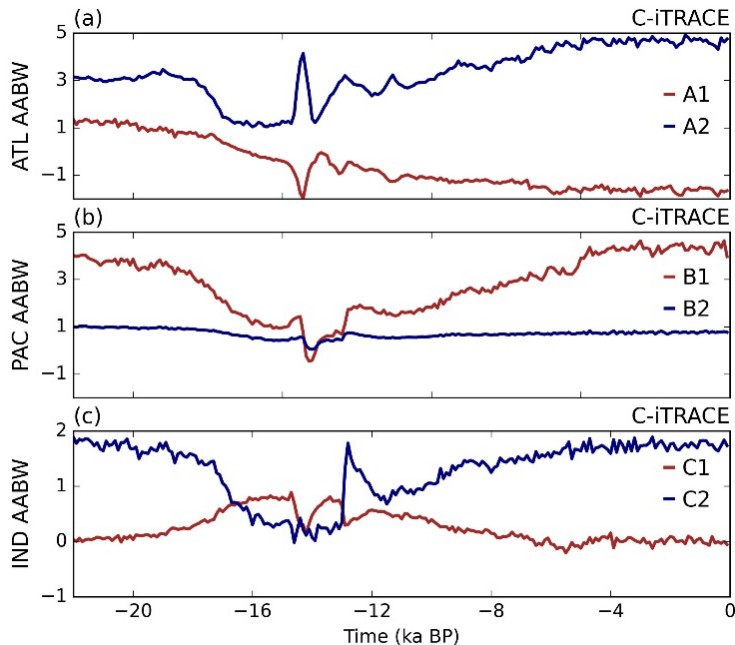

**Figure 8 Time evolutions of AABW export pathways into the (a) Atlantic, (b) Pacific, and (c) Indian Oceans in the C-iTRACE. Each branch of AABW transport (unit in Sv) across basins is calculated as the northward integrated volume transport within the black boxes on Figure 5a, 5e, and 5i, noted as A1 and A2 for the Atlantic, B1 and B2 for the Pacific, and C1 and C2 for Indian Ocean.**

## 4 iTRACE results

In addition to C-iTRACE, iTRACE simulation with a higher resolution is also explored. Results in the iTRACE and C-iTRACE display some similarities. The iTRACE simulation consistently simulates a younger ventilation age at the LGM compared to the PD. The IAGE averaged below 1 km is 626 years and 915 years at the LGM for the global mean and Pacific mean, in contrast to 2011 years and 3203 years at the PD, respectively (Fig. 9a-b). The younger LGM IAGE also appears to

be linked to the strong AABW in the iTRACE. The maximum magnitude in the abyssal GMOC is 11.6 Sv at the LGM and

5.6 Sv at the PD (Fig. 9d). Hence, the simulation results in both the C-iTRACE and iTRACE point toward a robust younger

ventilation age at the LGM than at the PD, primarily because of the faster overturning rate of AABW relative to the PD.

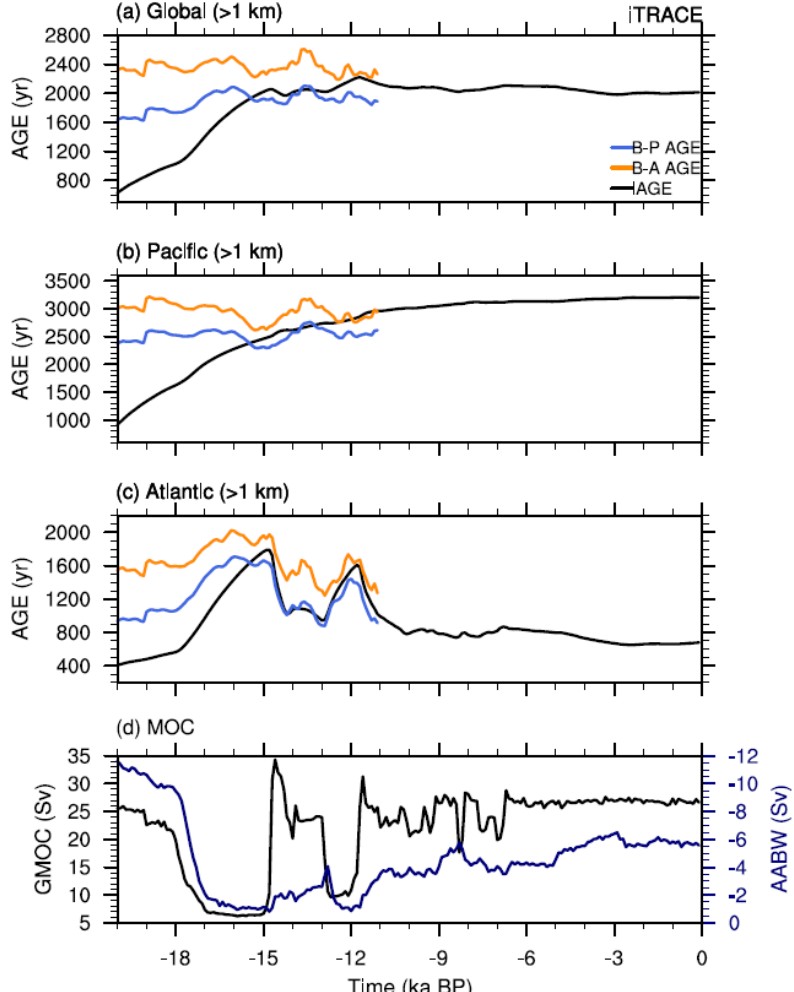

**Figure 9 Time evolutions in the iTRACE: (a) The global mean ideal age (IAGE; black), B-A age (yellow), and B-P age (blue) averaged below 1 km. (b) The Pacific mean IAGE (black), B-A age (yellow), and B-P age (blue) averaged below 1 km. (c) The**

**Atlantic mean IAGE (black), B-A age (yellow), and B-P age (blue) averaged below 1 km. (d) The GMOC (black) and AABW strength (navy). GMOC intensity is diagnosed as the maximum in the GMOC streamfunction below 500 m over 33° S-60° N, and AABW strength is diagnosed as the maximum in the GMOC streamfunction below 2km from 2° S-70° S.**

Moreover, the analysis of the deglacial ventilation age and AABW transport across the different basins in the iTRACE

supports results in the C-iTRACE. The IAGE averaged below 3.4 km increases to its maximum of 2460 years at 14.7 ka in

the Atlantic (Fig. 11e), and 3500 years at 6.6 ka in the Indo-Pacific basin (Fig. 12e). The older IAGE in the Pacific suggests

that the oldest water always occurs in the deep Pacific dominating the global ocean ventilation age during the last

deglaciation. The dominant role of Pacific on deglacial ventilation age appears to be unsurprising, as the Indo-Pacific sector

is only gradually ventilated by upwelled AABW water mass through diapycnal diffusion, resulting in ventilation occurring over the millennial timescale (Talley, 2013; Cessi, 2019; Ito and Marshall, 2008).

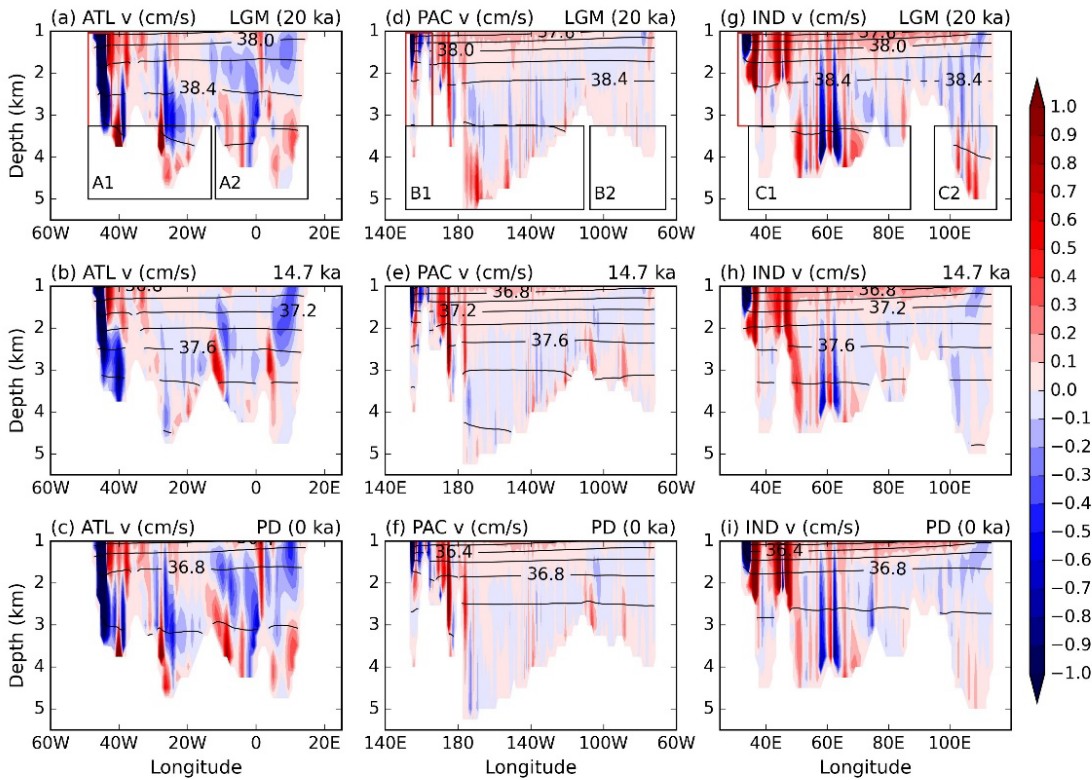

**Figure 10 Meridional velocity (shading) and potential density (lines) referenced to 2000 m at 30° S during the LGM (20 ka), 14.7 ka, and PD (0 ka) in the iTRACE. The shading contour interval is 0.1 cm/s and the black line contour interval is 0.2 kg/m³. The deep western boundary current excluding AABW is defined as the southward integrated volume transport within the highlighted red box on (a, d, g) for the Atlantic, Pacific, and Indian Oceans respectively. The export pathways of AABW in each basin is**
335 **calculated as the northward integrated volume transport within the black boxes on (a, d, g), noted as A1 and A2 for the Atlantic, B1 and B2 for the Pacific, and C1 and C2 for Indian Ocean.**

More quantitatively, the northward AABW DWBC transport in each basin is calculated from meridional velocity cross section at 30° S, using the same method as in the C-iTRACE. This is due to the approximately similar flat isopycnals below 3.4 km at 30° S in the iTRACE simulation (Fig. 10). The calculated southward DWBC and northward AABW DWBC are
340 consistent with the model MOCs, which are defined the same as in the C-iTRACE. The southward DWBC transport and upper cell are mainly differed by the northward return flow in the interior oceans (Fig. 11a-b, Fig. 12a-b). The calculated northward AABW DWBC transport corresponds well with the abyssal cell in each basin (Fig. 11c-d, Fig. 12c-d). Again, in the iTRACE, consistent results are obtained such that the correlation between IAGE and northward AABW transport is particularly evident in each basin. Specifically, in the Atlantic, the increasing of IAGE is in alignment with weakening of
345 AAWB transport (Fig. 11c and e). In the deep Indo-Pacific region, the IAGE increases from 825 years at the LGM to 3380

years at 11 ka. After 11 ka, the IAGE remains relatively old around 3400 years, corresponding to the continuous weak AABW transport (Fig. 12c and e). Overall, it suggests that weakening AABW transport are the main cause of the old ventilation ages during deglaciation.

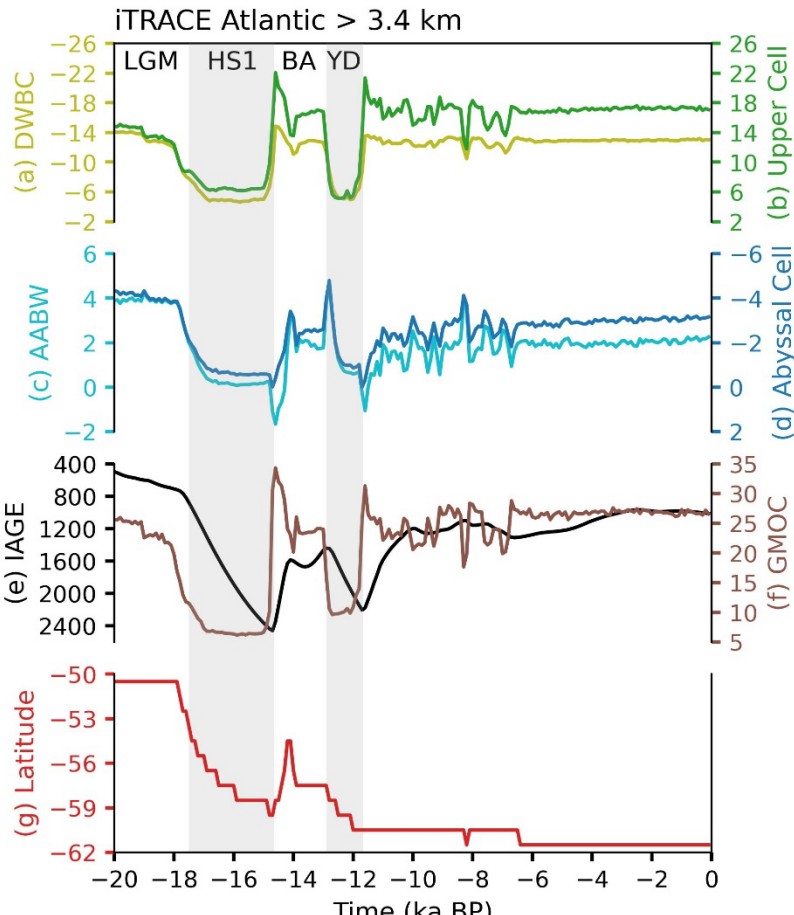

Figure 11 Atlantic deglacial evolutions in the iTRACE: (a) Southward DWBC (light olive, unit in Sv) defined as the southward integrated volume transport within the red box in Figure 10, (b) Upper cell (green, unit in Sv) defined as the maximum in AMOC between 0.6-3.5 km at 30° S, (c) AABW transport (cyan, unit in Sv) defined as $\Psi_{AABW} = -\int_{z=-H}^{z=-3.4km} \int v\,dxdz$ where v is meridional velocity component and H is the bottom seafloor, (d) abyssal cell (blue, unit in Sv) defined as the minimum in AMOC below 2.5 km m at 30° S, (e) IAGE (black, unit in yr) averaged below 3.4 km, (f) GMOC strength (brown, unit in Sv), and (g) the latitude of 20% sea ice coverage averaged over Southern Ocean (red). The gray shadings delimit periods of abrupt change. LGM-Last Glacial Maximum; HS1-Heinrich Stadial 1; BA-Bølling-Allerød; YD-Younger Dryas.

Physically, the water in the deep overturning regions gradually recirculates into the other ocean basins so that age generally increases with distance from the formation regions. The reduced AABW transport amounts to longer transit time from the formation site at the surface to the abyssal ocean, leading to the increased IAGE. The weakening of AABW is further suggested to be attributed to the surface buoyancy forcing over the Southern Ocean mainly in response to the deglacial atmospheric $CO_2$ increase and retreating ice sheets on land (Ferrari et al., 2014; Jansen, 2017; Jansen et al., 2018; Jansen and

Nadeau, 2016; Liu, 2023; Pedro et al., 2018). During the HS1, the freshwater input in the northern North Atlantic reduces NADW formation, leading to the slowdown in the Atlantic Meridional Overturning Circulation (AMOC) and reduced heat transfer into the North Atlantic. Consequently, heat accumulates in the Southern Hemisphere resulting in a warming in the Southern Ocean. Collectively, with the deglacial increase of atmospheric $CO_2$, sea ice around Antarctica is retreated with less brine rejection, ultimately contributing to the weakening of AABW transport towards north (Fig. 6i-j, Fig. 7i-j, Fig. 11g, and Fig. 12g).

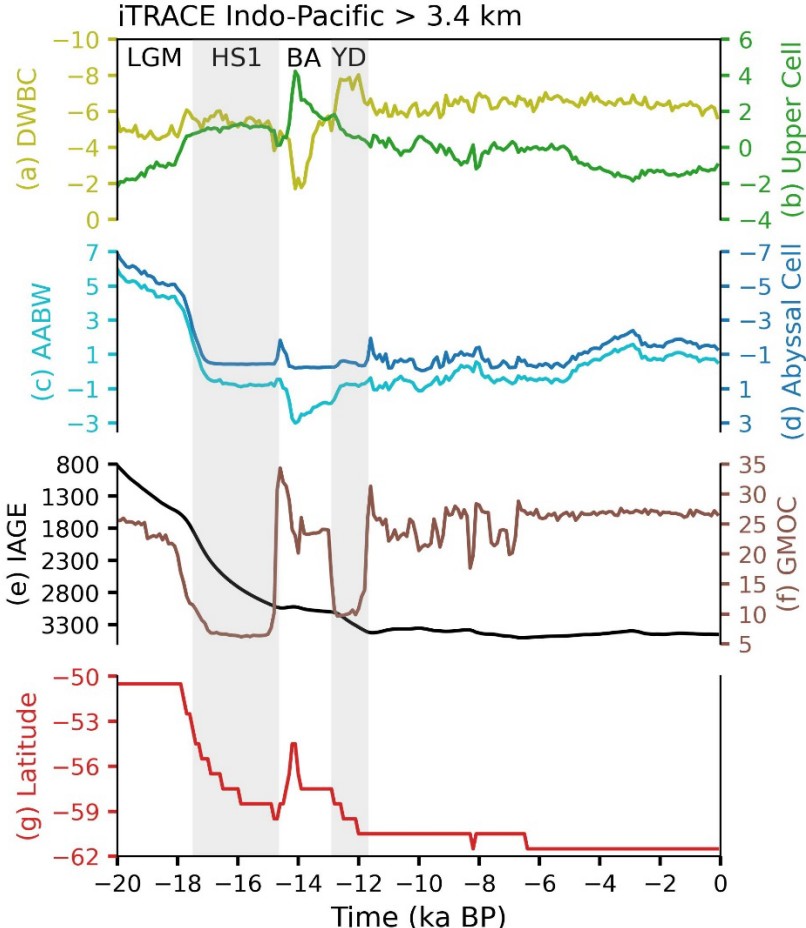

**Figure 12 Similar to Fig. 11 but for Indo-Pacific basin.**

It should be noted that, in comparison with the bell-shaped temporal pattern observed in the C-iTRACE, the deglacial evolution of IAGE exhibits a distinct temporal behaviour in the iTRACE, with the global deep IAGE increasing from the LGM to the HS1, followed by a continuous presence of old ventilation age until the PD (Fig. 9a). A similar pattern is also seen for the Pacific mean (Fig. 9b). Nevertheless, this distinct deglacial evolution of IAGE is coupled with the transport evolution of AABW (Fig. 9d). A strong AABW transport is simulated at the LGM, followed by a significant reduction during the HS1 and small variability during the Bølling-Allerød (BA, 14.7-12.9 ka) and YD intervals. After the Holocene,

the AABW transport slowly recovers to its present-day value, which is about half (52%) smaller than the LGM value. Consequently, the deep ocean is ventilated at a slower rate due to weaker AABW transport during the Holocene, resulting in a continuous presence of old ventilation age. This is obviously different from C-iTRACE. In the C-iTRACE, the AABW transport at the PD is only 20% smaller than that at the LGM. This in turn results in a slightly older ventilation age at the PD

relative to the LGM in the C-iTRACE. Furthermore, the export pathways of AABW from the Southern Ocean into each basin in the iTRACE is somewhat distinct from C-iTRACE results. The major difference between the two simulations lies in the main export route in the Atlantic. In the iTRACE, AABW tends to flow northward into the Atlantic along the west of the Mid Atlantic Ridge in the Atlantic Ocean (Fig. 13a); along the southwest Pacific basin (Fig. 13b); and along the eastern Indian Ocean basin (Fig. 13c). Nonetheless, the correlation between deep ocean ventilation age and AABW intensity are

clearly robust in both simulations.

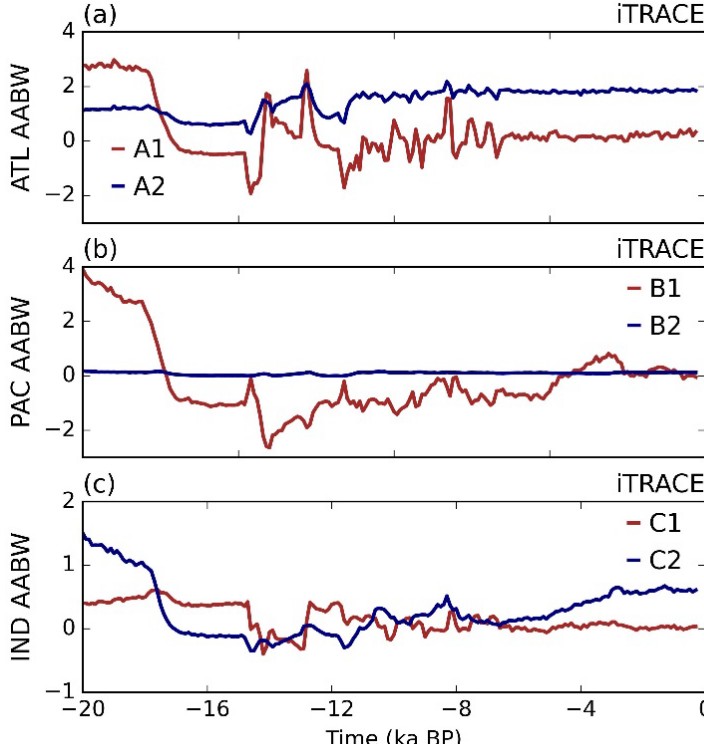

**Figure 13 Similar to Fig. 8 but in the iTRACE (unit in Sv).**

**5 Summary**

In this study, mechanisms of global ventilation age changes during the last deglaciation have been assessed using the ideal

age tracer in two deglacial simulations, C-iTRACE and iTRACE. Results from both simulations suggest that, in contrast to the radiocarbon ventilation ages showing poorly ventilated glacial deep waters, the true model ventilation age (IAGE)

exhibits an overall modestly younger water age during the LGM relative to the PD, with global mean of 800 (630) years and 930 (2000) years at the LGM and present day, respectively, in two simulations, mainly caused by the faster overturning of AABW. This is also inferred from the new approach of estimated deep ocean water age using radiocarbon age by considering multiple water mass contributions (Fig. 1). This implies that radiocarbon reconstructions are likely biased and may fail to reproduce the true deep ocean ventilation age at the LGM. More importantly, the important role of AABW transport in shaping the global true oceanic ventilation age during deglaciation are confirmed robustly across two transient simulations. The oldest water always occurs around in the deep Pacific, dominating global ocean ventilation age. The global mean ventilation ages peak at 1900 and 2200 years around 14-12 ka in the simulations. And the increase of deglacial global deep ocean ventilation age is highly correlated with the reduced transport of the AABW associated with the reduced sea ice coverage and negative buoyancy flux over the Southern Ocean.

However, it should be noted that there are some differences between these two simulations. Major difference is the distinct evolution of AABW volume transport, particularly in the Indo-Pacific sector. In the iTRACE, the AABW transport is simulated to be weaker during the Holocene compared to the C-iTRACE, resulting in a continuous presence of old water age from the late HS1 to early Holocene. This continuous old IAGE during the Holocene is consistent with a recent study showing high-precision radiocarbon results from the Southern Ocean and North Atlantic, demonstrating the millennial stability of Holocene overturning circulation (Chen et al., 2023). In this case, results from iTRACE with a higher resolution seems to be more realistic, showing that the ventilation age during the LGM can be much younger than the age at the PD. Therefore, studies with multi-model intercomparisons are highly desirable to validate our findings, and increased proxy observations would greatly contribute to further constraining the changes in deglacial ventilation age in the ocean reservoir.

### Data Availability

The C-iTRACE data used in this study is available through the NCAR/UCAR Digital Asset Services Hub at https://gdex.ucar.edu/dataset/204_ajahn.html (Gu et al., 2021a). The iTRACE data used in the study are publicly available at https://www.earthsystemgrid.org/dataset/ucar.cgd.ccsm4.iTRACE.html.

### Author contribution

**Lingwei Li**: Conceptualization, Formal analysis, Investigation, Methodology, Writing - original draft preparation. **Zhengyu Liu**: Conceptualization, Funding acquisition, Methodology, Writing - review & editing. **Jinbo Du**: Conceptualization, Writing - review & editing. **Lingfeng Wan**: Writing - review & editing. **Jiuyou Lu**: Writing - review & editing

Competing interests

The authors declare that they have no conflict of interest.

**Acknowledgments**

This project is supported by NSF (OCN1810681), the national key research and development program (2023YFC3107701), National Natural Science Foundation of China Key program (42130604), Science and Technology Innovation Project of Laoshan Laboratory (No. LSKJ202203303, LSKJ202202201, LSKJ202300401), and Fundamental Research Funds for the

Central Universities (202362001, 202072010). The CESM project is supported primarily by the NSF. This material is based on study supported by the National Center for Atmospheric Research, which is a major facility sponsored by the NSF under Cooperative Agreement no. 1852977. Computing and data storage resources, including the Cheyenne supercomputer doi:10.5065/D6RX99HX), were provided by the Computational and Information Systems Laboratory (CISL) at NCAR.

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
