# Peer review of "Mechanisms of Global Ocean Ventilation Age Change during the Last Deglaciation"

_EGUsphere, 2023_

## Author Comment (AC1)

Dear Editor and reviewers,
We appreciate the expert comments provided by reviewers. Below we provide a detailed point-by-point response to all these comments. The reviewers' comments are shown in black, and our responses follow in red font. The quoted texts in the revised manuscript are in blue. Line numbers in this response reflect those in the revised manuscript unless otherwise noted. We thank the reviewers for the detailed comments and suggestions, also the time and effort that they invested into the review. We believe the revisions have strengthened the manuscript.

General Comments
1. The manuscript lacks a detailed description of ideal age tracer in these models, as readers in the community may not be familiar with C-iTRACE and iTRACE modeling. Therefore, it is necessary to provide a clearer context on model set up for idea age tracer (e.g. England, 1995). Thanks for the suggestions. This is now clarified in revised manuscript as "First, the ideal age (IAGE) is included, which is set to 0 at the ocean surface and ages at a rate of 1 year/year thereafter passively advected and diffused into the ocean interior. Thus, IAGE is a passive circulation tracer measuring the time elapsed since the last contact with the atmosphere and working like a clock counting time after being restored to zero (England, 1995; Koeve et al., 2015). In turn, the IAGE represents the "true" model ventilation age, although it does not account for the insulation effect of sea ice" in line 109-113.

2. It would be worthwhile to discuss the importance of understanding the age of water and ventilation in the introduction.
The introduction in the revised manuscript now reads: "Discussion about past and future climate change is often difficult without reference to the global ocean circulation, because of its critical role in storing and transporting heat, carbon, and nutrients. Ice core records demonstrate significant variability in atmospheric carbon dioxide ($CO_2$) concentrations on the glacial-interglacial timescale (Sigman and Boyle, 2000; Monnin et al., 2001; Schmitt et al., 2012). During the last glacial period, atmospheric $CO_2$ levels were approximately 90 parts per million by volume (ppmv) lower than the Holocene value of 280 ppmv (11.7-0 ka where ka indicates "thousand years ago). This variation in $CO_2$ is closely coupled with long-term climate changes and the carbon cycle in the Earth system. Various oceanic processes thus have been suggested to regulate atmospheric $CO_2$, such as air-sea gas exchange (Long et al., 2021), biological production (Broecker, 1982; Sigman et al., 2021; Sigman and Boyle, 2000), and ocean circulation (Ai et al., 2020; Marcott et al., 2014; Tschumi et al., 2011; Skinner et al., 2015, 2010). Consequently, understanding past changes in global ocean circulation provides a good opportunity, not only for a more accurate constraint on the past climate change and carbon cycle, but also for insight into future climate change with rising atmospheric $CO_2$ level" in line 21-31.

3. It would be worthwhile to describe how the strength of the AABW physically affects the ventilation processes in two basins and, as AABW is part of the AMOC, why the AABW transport reduces during the deglaciation from a coupled view.
This is clarified in the revised manuscript as "Physically, the water in the deep overturning regions gradually recirculates into the other ocean basins so that age generally increases with distance from the formation regions. The reduced AABW transport amounts to longer transit time from the formation site at the surface to the abyssal ocean, leading to the increased IAGE. The weakening of AABW is further suggested to be attributed to the surface buoyancy forcing

over the Southern Ocean mainly in response to the deglacial atmospheric $CO_2$ increase and retreating ice sheets on land (Ferrari et al., 2014; Jansen, 2017; Jansen et al., 2018; Jansen and Nadeau, 2016; Liu, 2023; Pedro et al., 2018). During the HS1, the freshwater input in the northern North Atlantic reduces NADW formation, leading to the slowdown in the Atlantic Meridional Overturning Circulation (AMOC) and reduced heat transfer into the North Atlantic. Consequently, heat accumulates in the Southern Hemisphere resulting in a warming in the Southern Ocean. Collectively, with the deglacial increase of atmospheric $CO_2$, sea ice around Antarctica is retreated with less brine rejection, ultimately contributing to the weakening of AABW transport towards north (Fig. 6i-j, Fig. 7i-j, Fig. 11g, and Fig. 12g)" in line 350-360.

4. Figure 4: What is driving the IAGE from the North Atlantic to get much older from 12 to 10 ka?
This is clarified in revised manuscript as "The mean IAGE for AABW water mass increases from 836 years at the LGM to 1813 years at 14 ka due to the weakening of AABW transport, while the age of NADW increases by up to 1500 years during the period of 12–11 ka due to more NADW sinking into deeper depths in the Arctic (Fig. 2j-l), resulting in a relatively older water age for NADW and the lag of 2000 years between the maxima of Dye_NA and Dye_S (Fig. 3a-b)" in line 272-275.

5. No units in figures 6, 7, 8, 11, 12, and 13
Corrected.

6. Line 76 and 39: Check grammar
Corrected.

7. Line 8: Remove "postal code"
Corrected.

8. Line 11: "CO2" to "$CO_2$"
Corrected.

9. Line 95 and Line 368: "Gu, Liu, Jahn, et al., 2019", Remove "Liu, Jahn"
Corrected.

10. What's the meaning in Line 170: "8 $10^8$ km$^3$" and "7 $10^{11}$ yr km$^3$".
The is corrected in the revised manuscript: "Furthermore, the volume of AABW is about $9\times10^8$ km$^3$ at the LGM and PD, such that the volume integrated IAGE for AABW is about $7\times10^{11}$ yr km$^3$ at both LGM and PD (Fig. 3b and 3c)" in line 189-191.

11. Line 45: Remove the comma symbol in "Zanowski et al., (2022)"
Corrected.

---

## Author Comment (AC2)

Dear Editor and reviewers,

We appreciate the expert comments provided by reviewers. Below we provide a detailed point-by-point response to all these comments. The reviewers' comments are shown in black, and our responses follow in red font. The quoted texts in the revised manuscript are in blue. Line numbers in this response reflect those in the revised manuscript unless otherwise noted. We thank the reviewers for the detailed comments and suggestions, also the time and effort that they invested into the review. We believe the revisions have strengthened the manuscript.

Major comments:

The "Introduction" section should be reorganized. The topic of the article is the underlying mechanisms for the younger ventilation during the LGM relative to the PD, in tandem with the oldest ventilation age during the last deglacial according to the IAGE. However, the present Introduction introduces too much about the $\Delta^{14}C$ age, and the definition and application of the IAGE are lacking. Moreover, the significance of understanding the ventilation age during the LGM and last deglacial should be highlighted more clearly.

Thanks for the comments. This manuscript is actually motivated by the different temporal evolution of $\Delta^{14}C$ age and ideal age. Therefore, it is our opinion that a brief introduction about the $\Delta^{14}C$ age is necessary, and a more detailed definition of IAGE is elaborated in section 2. The introduction is revised in the manuscript to highlight the importance of ventilation age: "Discussion about past and future climate change is often difficult without reference to the global ocean circulation, because of its critical role in storing and transporting heat, carbon, and nutrients. Ice core records demonstrate significant variability in atmospheric carbon dioxide ($CO_2$) concentrations on the glacial-interglacial timescale (Sigman and Boyle, 2000; Monnin et al., 2001; Schmitt et al., 2012). During the last glacial period, atmospheric $CO_2$ levels were approximately 90 parts per million by volume (ppmv) lower than the Holocene value of 280 ppmv (11.7-0 ka where ka indicates "thousand years ago). This variation in $CO_2$ is closely coupled with long-term climate changes and the carbon cycle in the Earth system. Various oceanic processes thus have been suggested to regulate atmospheric $CO_2$, such as air-sea gas exchange (Long et al., 2021), biological production (Broecker, 1982; Sigman et al., 2021; Sigman and Boyle, 2000), and ocean circulation (Ai et al., 2020; Marcott et al., 2014; Tschumi et al., 2011; Skinner et al., 2015, 2010). Consequently, understanding past changes in global ocean circulation provides a good opportunity, not only for a more accurate constraint on the past climate change and carbon cycle, but also for insight into future climate change with rising atmospheric $CO_2$ level" in line 21-31, and details of IAGE set up is now clarified in the revised manuscript: "First, the ideal age (IAGE) is included, which is set to 0 at the ocean surface and ages at a rate of 1 year/year thereafter passively advected and diffused into the ocean interior. Thus, IAGE is a passive circulation tracer measuring the time elapsed since the last contact with the atmosphere and working like a clock counting time after being restored to zero (England, 1995; Koeve et al., 2015). In turn, the IAGE represents the "true" model ventilation age, although it does not account for the insulation effect of sea ice" in line 109-113.

2. Line 168: How is the threshold of 70% defined? It is not clarified in the manuscript. Whether the evolution results would be different if using different values (such as 60%, 80%)?

The other reviewer brought up a similar concern which is now addressed. We have performed additional analysis showing that the conclusion is actually somewhat insensitive to the choice of this value, which is now clarified in the revised manuscript: "The IAGE value is calculated

where the percentages of AABW% and NADW% exceed 70% below 1 km in the global ocean (Fig. 3a). The conclusion is somewhat insensitive to the choice of dye concentration value for NADW and AABW. A higher (lower) value of 80% (60%) indicates less (more) mixing with other water masses and thus does not affect the main results" in line 185-188. Figure R1 below shows the results using 60%, and the results are similar to that using the fraction value of 70% in figure 3 in the manuscript.

[Figure]

Figure R1 Time evolutions for water masses sourced from the North Atlantic (Dye_NA) and Southern Ocean (Dye_S). (a) The global volume-weighted average of ideal age (IAGE) where fractions of water are greater than 60%. (b) The total water volume where fractions of water are greater than 60%. (c) The age of total volume of water from the Southern Ocean and North Atlantic.

3. Line 190-195: Authors indicated that the evolution of IAGE is more similar to the strength of AABW than GMOC. However, changes in the AABW transport may be a part of GMOC. In other words, the AABW transport and GMOC are interactive, and in the modelling who is the reason and who is the result? It makes me confused about the appearance of the GMOC across the manuscript. More discussions or clarifications about this point should be added.
We thanks for the comments, and this is now clarified in the revised manuscript: "This bell shaped deglacial evolution of global IAGE aligns with similar transport evolutions of the residual GMOC and global AABW (Fig. 1d, Fig. 2d-f) as well as the AABW% in dye tracer from the Southern Ocean (Fig. 3, Fig. 2g-i). Here, the GMOC is diagnosed as the maximum in the GMOC streamfunction below 600 m from 33° S-60° N and, therefore, the main feature of the GMOC follows the upper clockwise cell mostly confined to the Atlantic sector. Since the abyssal ocean is predominately filled by AABW, the evolution of IAGE corresponds more closely to the

AABW intensity than GMOC as shown in figure 1 that the decrease in global AABW transport aligns with the increase in global mean IAGE during the same period (14–13 ka)" in line 203-210.

4. Authors highlighted the importance of the AABW transport changes in regulating the ventilation age during the LGM and the last deglacial and indicated that the AABW transport changes are associated with sea ice and buoyancy flux over the Southern Ocean. However, this conclusion only appears simply in the Abstract and Summary, and analysis is lacking in the manuscript. More discussions about the mechanisms driving AABW transports should be added. At least, the evolution of sea ice during the LGM and last deglacial should be provided and the relationship between sea ice and AABW transport needs to be analyzed briefly. Moreover, the ultimate driving factors are also necessary to be discussed (i.e. the external forcings), maybe the role of freshwater injection or continental ice sheet during the LGM and last deglacial on the sea ice, on the AABW transport, and on the ventilation age should be discussed.

The evolution of sea ice, as suggested, is added in figure 6, 7, 11, and 12. And the role of freshwater injection and continental ice sheet in AABW and ventilation age is also discussed. The revised manuscript now reads: "Model simulations further suggest that the strong glacial AABW transport is dominated by changes in surface buoyancy forcing over the Southern Ocean (Ferrari et al., 2014; Jansen, 2017; Jansen and Nadeau, 2016; Liu, 2023; Shin et al., 2003; Sun et al., 2018). The sea ice expansion at the LGM enhanced the brine rejection during winter, leading to extremely saline and dense AABW. Thus, the glacial deep ocean is filled with greatly expanded cold-salty AABW water mass with higher densities, contributing to a stronger AABW transport. The stronger glacial AABW transport, in turn, reflects a shorter residence time in the ocean interior, as indicated by the overall younger LGM IAGE compared to the water age at the PD" in line 161-167, and "Physically, the water in the deep overturning regions gradually recirculates into the other ocean basins so that age generally increases with distance from the formation regions. The reduced AABW transport amounts to longer transit time from the formation site at the surface to the abyssal ocean, leading to the increased IAGE. The weakening of AABW is further suggested to be attributed to the surface buoyancy forcing over the Southern Ocean mainly in response to the deglacial atmospheric $CO_2$ increase and retreating ice sheets on land (Ferrari et al., 2014; Jansen, 2017; Jansen et al., 2018; Jansen and Nadeau, 2016; Liu, 2023; Pedro et al., 2018). During the HS1, the freshwater input in the northern North Atlantic reduces NADW formation, leading to the slowdown in the Atlantic Meridional Overturning Circulation (AMOC) and reduced heat transfer into the North Atlantic. Consequently, heat accumulates in the Southern Hemisphere resulting in a warming in the Southern Ocean. Collectively, with the deglacial increase of atmospheric $CO_2$, sea ice around Antarctica is retreated with less brine rejection, ultimately contributing to the weakening of AABW transport towards north (Fig. 6i-j, Fig. 7i-j, Fig. 11g, and Fig. 12g)" in line 350-360.

5. The simulated ventilation age during the LGM and last deglacial need to be compared with the various proxy reconstructions comprehensively in the manuscript. Please add some discussions on this point.

We acknowledge the importance of model validation using proxy observations. However, the transient ocean simulation of the last deglaciation (C-iTRACE) has been examined in greater detail in Gu et al. (2020, 2021), Zanowski et al. (2022). The scope of this modeling study is to better understand the deglacial evolution of deep ocean ventilation age.

Minor comments:

1. Fig. 1 only provides the global mean IAGE and Pacific mean IAGE. What is about the Atlantic mean? Any opinion?

Overall, the water age in the Atlantic is considerably younger than that in the Pacific. As a result, Atlantic was not included in Figure 1 in the original manuscript. But it is now included in figure 1 in revised manuscript.

2. Line 124: suggest replacing "in contrast to" with "younger than"

Revised as suggested.

3. Lines 126-127: This sentence is confusing. Pleas rewrite.

Revised as suggested.

4. Lines 152-154: The description here is unexpected, and I suggest removing it or making it clear.

Revised as suggested.

5. Line 170: The multiple sign is missing.

Corrected.

6. The order of subpanels in Fig. 6 and Fig. 7 should be rearranged, as Fig.6g-h and Fig. 7g-h appear earlier than Fig. 6a and 7a. (Line 244-245).

Corrected.

7. Fig. 6 and Fig. 7. The left string of figures may be "C-iTRACE".

Corrected.

7. More quantified information should be added in the Abstract and Summary section (for example, the exact value of the ventilation age during the LMG and the last deglacial).

Revised as suggested.

---

## Author Comment (AC3)

Dear Editor and reviewers,

We appreciate the expert comments provided by reviewers. Below we provide a detailed point-by-point response to all these comments. The reviewers' comments are shown in black, and our responses follow in red font. The quoted texts in the revised manuscript are in blue. Line numbers in this response reflect those in the revised manuscript unless otherwise noted. We thank the reviewers for the detailed comments and suggestions, also the time and effort that they invested into the review. We believe the revisions have strengthened the manuscript.

General Comments

The paper is motivated by the differences between the temporal evolution of $^{14}$C ages and ideal ages shown in Figures 1 and 9. That $^{14}$C ages are higher than ideal ages is not new (e.g. Koeve et al. 2015). There are also other models showing for the LGM higher $^{14}$C ages but at the same time younger ideal ages compared to PD (e.g. Galbraith and de Lavergne 2019; see also the discussion by Skinner and Bard 2022). Apart from the introduction, $^{14}$C ages are not really discussed or compared with observations. In fact, this has been done by other authors (Gu et al. 2020, Zanowski et al. 2022). The focus of Li et al. is on the discussion of simulated ideal ages. Here, a systematic problem is that there is no way to validate ideal ages with observations from the past. Therefore, the results by Li et al. may be helpful to understand the model behaviour in the C-iTRACE and iTRACE simulations, but the added value for our understanding of real marine proxy records is limited.

Respectfully, we do not agree with this comment. The scope of this modeling study is to better understand the simulated deglacial evolution of deep ocean ventilation age, focusing on the feedback mechanisms in one of the most widely used Earth system models. The focus is not model-proxy comparison, because the transient ocean simulation of the last deglaciation (C-iTRACE) have been examined in detail in Gu et al. (2020, 2021b) and Zanowski et al. (2022).

Specific comments

Figure 1 and line 75: "incredibly identical" "BwP" ages. This is indeed incredible unless further details of this unpublished approach are provided.

We thank this reviewer for the language suggestion. In the revised manuscript "incredible" is removed and this sentence now reads "This new approach of BwP age is considerably similar to the true ocean ventilation age globally" in line 82-83.

Figure 1: "BP" (= 1950 CE) should be defined somewhere (maybe near line 23)

This is clarified in revised manuscript as "all times (calendar ages) are reported in thousands of years before present (ka BP where BP indicates 1950 CE)" in line 130.

Figures 1 and 9: It would be worthwhile to include the Atlantic to facilitate the understanding of Figs. 6 and 8

Revised as suggested.

Line 56: "AABW is defined as the minimum (...) from 2°S-70°S" – does it make sense in this context to consider 2°S which is far away from the source water = ventilation regions?

The AABW transport in the manuscript is diagnosed as the minimum value in the Global Meridional Overturning Circulation (GMOC) streamfunction below 2 km. Therefore, it is reasonable to consider the abyssal ocean as far as 2° S to define overall global AABW transport.

This is clarified in the revised figure captions: "Here the GMOC intensity is diagnosed as the maximum in the GMOC streamfunction below 600 m from 33° S-60° N, and AABW is diagnosed as the minimum in the GMOC streamfunction below 2 km over 2° S-70° S".

Line 57: "ventilation time" should read "ventilation age"
Corrected.

Lines 61-62: "both the B-A and B-P age are remarkably old[er] at the LGM than the present day". This is not really the case for B-P ages shown in Fig. 1.
As shown in Figure 1, in the C-iTRACE simulation, the B-P age is approximately 1600 and 1480 years at the LGM and PD, respectively. This is clarified in revised manuscript: "First, both the B-A age and B-P age are considerably older at the LGM (~20 ka) than the preindustrial period (denoted here as the present day or PD, ~0 ka)" in line 60-61.

Line 79: There are no "true ventilation" proxies, see my comment above
We agree. In principle, radiocarbons have been considered as the best proxy for estimating the water age. However, it is still challenging to have an accurate estimation of deep ocean water age from radiocarbon (e.g. due to marine reservoir age). Therefore, the purpose of this manuscript is to provide additional model perspectives on deglacial changes in the deep ocean ventilation age and associated mechanisms. This is clarified in the manuscript: "Due to the potential challenges in radiocarbon proxies to accurately estimate the deep ocean ventilation age, we will address the second question regarding the mechanism of the deglacial evolution of the model ventilation time, in order to provide additional model perspectives on changes in the global ocean ventilation age during the last deglaciation" in line 86-89.

Line 120: "Ventilation" originally meant "oxygenation" of the deep sea; in this sense "ventilation ages" and "ideal ages" are not really the same for (chemical) oceanographers.
Indeed, these are different concepts. However, it is also common practice in modeling studies to track the water ventilation age using passive ideal age tracers, see England (1995), Galbraith and de Lavergne (2019), etc.

Figure 2 (h)-(i), lines 133 and 168: Contour lines in Fig. 2 represent the value of 0.8 but ideal ages are calculated where the percentages of AABW and NADW exceed 70%. This should be consistent (i.e., 0.8/80% or 0.7/70%)
We thank this reviewer for the suggestion. The contour lines in figure 2 are now corrected to represent the value of 0.7.

Figure 3 (a): What is the reason of the lag of ~2 kyears between the maxima of DYE_NA and DYE_S?
The maxima of DYE_S is due to the weakening of AABW transport, and the maxima of DYE_NA is due to NADW water masses sinking into deeper depth in the Arctic region. This is now clarified in the revised manuscript "The mean IAGE for AABW water mass increases from 836 years at the LGM to 1813 years at 14 ka due to the weakening of AABW transport, while the age of NADW increases by up to 1500 years during the period of 12–11 ka due to more NADW sinking into deeper depths in the Arctic (Fig. 2j-l), resulting in a relatively older water age for

NADW and the lag of 2000 years between the maxima of Dye_NA and Dye_S (Fig. 3a-b)" in line 274-277.

Line 193: "decrease in AABW transport and increase in IAGE during the same period (14-13 ka)" is at odds with Figure 1
This statement is actually consistent with Figure 1. Between 14 and 13 ka, the decrease in the global AABW transport (blue line in Figure 1D) corresponds to an increase in the global mean IAGE (black line in Figure 1A). We see how our original language could be confusing, which is now clarified in the revised manuscript: "This bell shaped deglacial evolution of global IAGE aligns with similar transport evolutions of the residual GMOC and global AABW (Fig. 1d, Fig. 2d-f) as well as the AABW% in dye tracer from the Southern Ocean (Fig. 3, Fig. 2g-i). Here, the GMOC is diagnosed as the maximum in the GMOC streamfunction below 600 m from 33° S-60° N and, therefore, the main feature of the GMOC follows the upper clockwise cell mostly confined to the Atlantic sector. Since the abyssal ocean is predominately filled by AABW, the evolution of IAGE corresponds more closely to the AABW intensity than GMOC as shown in figure 1 that the decrease in global AABW transport aligns with the increase in global mean IAGE during the same period (14–13 ka)" in line 203-210.

[Figure]

Figure R1 Time evolutions in the C-iTRACE: (a) The global mean ideal age (IAGE; black), benthic-atmosphere $\Delta^{14}$C age (B-A age; yellow), benthic-planktonic $\Delta^{14}$C age (B-P age; blue), and weighted benthic-planktonic $\Delta^{14}$C age (BwP age; green) averaged below 1 km. (b) The Pacific mean IAGE (black), B-A age (yellow), B-P age (blue), and BwP age (green) averaged below 1 km. (c) The Atlantic mean IAGE (black), B-A age (yellow), B-P age (blue), and BwP

age (green) averaged below 1 km. (d) The Global Meridional Overturning Streamfunction (GMOC; black) and Antarctic Bottom Water strength (AABW; navy). Here the GMOC intensity is diagnosed as the maximum in the GMOC streamfunction below 500 m from 33° S-60° N, and AABW is diagnosed as the minimum in the GMOC streamfunction below 2 km over 2° S-70° S.

Figure 4: The changes would become more obvious if the ages were scaled to values at 0 ka BP (i.e., if age anomalies were shown)
Revised as suggested.

Figure 4 / line 199: As the Pacific is connected with the Indian Ocean south of the equator, the Indo-Pacific MOC should be considered instead of the PMOC
Revised as suggested.

Figure 6, 7, 8, 11, 12, 13: Units are missing
Corrected.

Line 235-236: "isopycnals (...) at 30°S exhibit minimal changes below 3.4 km" – this is at odds with what can be seen in Figs. 2 and 5
The calculation of AABW DWBC actually has nothing to do with Figure 2. Contours in figure 2 are zonal mean potential density, while contours in figure 5 are zonal potential density at 30° S which are essentially all flat below 3.4 km. Therefore, the AABW transport are calculated at 30° S using the basin wide integrated volume transport below 3.4 km.

Line 242: "The calculated northward AABW DWBC transport aligns exceptionally well the transport of model abyssal upper cell in each basin (Fig. 6g-h, Fig. 7g-h)". Do you mean "abyssal cell", "upper cell" or "abyssal and upper cell"? I don't get that from Figs. 6 and 7.
Indeed. This is now clarified in revised manuscript: "The calculated northward AABW DWBC transport aligns exceptionally well with the transport of model abyssal cell in each basin" in 259-260.

Line 255: Why is the (ideal) ventilation age of AABW typically much older than the age of NADW?
This is now elaborated in the revised manuscript: "Note that the ventilation age of AABW water mass is typically much older than the age of NADW (Fig. 3). This is because the NADW water mass characterizes the southward branch of the upper cell confined to the Atlantic sector, which is interconnected with AABW and thermocline ventilation, leading to relatively short residence times through the ocean interior" in line 271-274.

Line 258-259: "more NADW sinking into deeper depths in the Arctic" – this is not in line with Fig. 3 (b) where the volume of DYE_NA remains almost constant.
There is actually a slight increase of volume of Dye_NA in figure 3b, although the volume of NADW is about 8 times smaller than the volume of AABW. Figure R2 shows that the global concentration of Dye_NA at 2 km during the 20ka, 16 ka, 14.5 ka, 12.9 ka, and 0ka, respectively. It clearly shows more NADW water mass in the Arctic since 14.5 ka and therefore the ventilation age of NADW gets older during the period of 14.5-10 ka in figure 3a.

[Figure]

Figure R2 Global distribution of Dye_NA concentrations at 20 ka, 16 ka, 14.5 ka, 12.9 ka, and 0 ka, respectively.

Line 315: "The calculated southward DWBC and northward AABW DWBC are validated by the model MOCs" – this is a tautology
The model MOCs here are direct diagnostic output from simulations, and by comparing the calculated transport of DWBCs using meridional velocity with model MOCs, we are able to access the accuracy and reliability of the calculations as well as main temporal features of DWBCs transports. Therefore, it is our opinion that it is necessary to verify the calculated DWBC with the model MOCs.

Line 349: See my comment concerning line 75
Revised as suggested. Reference is added in the revised manuscript.

References:
England, M. H. (1995). The age of water and ventilation timescales in a global ocean. *Journal of Physical Oceanography*, *25*, 2756–2777.

---

## Referee Report (RR1)

The manuscript has been improved but there are still issues which should be amended before I could recommend its publication:

L60-61: *"First, both the B-A age and B-P age are considerably older at the LGM (~20 ka) than the preindustrial period"*
"Considerably" is rather vague and should be quantified or be removed.

L82: *"This new approach of BwP age is considerably similar to the true ocean ventilation age globally"*
(1) The new approach is still unpublished, and Fig. 1 only shows that its results (but not that the various methods) are similar.
(2) "Considerably" is rather vague and should be quantified or be removed.
(3) The "true" ocean ventilation age is the "ideal" model age (IAGE).
Therefore, it would be more appropriate to rephrase, e.g., "These new BwP ages are similar to the ideal ocean ventilation tracer ages ... "

L194: *"dramatic bell shape"*
"Dramatic" should be removed.

L195-196: *"A similar bell shape pattern is also seen for the Pacific and Atlantic mean"*
There is definitely no bell shape in the Atlantic but a clear age drop near 14 ka BP.

L209-210: *" (...) the decrease in global AABW transport aligns with the increase in global mean IAGE during the same period (14–13 ka)."*
According to Fig. 1d, AABW abruptly increases between 14-13 ka BP, that is, before the age peak at 12.9 ka BP. What is the explanation for this lag?

L 274-277: *"The mean IAGE for AABW water mass increases from 836 years at the LGM to 1813 years at 14 ka due to the weakening of AABW transport, while the age of NADW increases by up to 1500 years during the period of 12–11 ka due to more NADW sinking into deeper depths in the Arctic (Fig. 2j-l), resulting in a relatively older water age for NADW and the lag of 2000 years between the maxima of Dye_NA and Dye_S (Fig. 3a-b)"*.
(1) It would be more obvious to show a time series of NADW strength which could be added to Fig 1d.
(2) How is it possible that enhanced NADW production, i.e. enhanced ocean ventilation can lead to water mass ageing?
(3) According to Fig 3b the volume of North Atlantic waters decreases between 14-12 ka. This seems to contradict more NADW sinking.
(4) I am not sure of how to interpret Fig. R3 in the author's reply to my first review. What are the units? It appears that DYE_NA at 2 km declines before the maximum of IAGE in Fig. 3a in the paper.

L333-334 (and L315 in the initial submission): *"The calculated southward DWBC and northward AABW DWBC are validated by the model MOCs, which are defined the same as in the C-iTRACE"*
You cannot validate model results with model results obtained with the same model and in the same simulation. If the results were not consistent something would be wrong.

---

## Author Response (AR2)

Dear Editor and reviewers,

We appreciate the expert comments provided by the reviewer. Below we provide a detailed point-by-point response to all these comments. The reviewer's comments are shown in black, and our responses follow in red font. The quoted texts in the revised manuscript are in blue. Line numbers in this response reflect those in the revised manuscript unless otherwise noted.

The manuscript has been improved but there are still issues which should be amended before I could recommend its publication:

L60-61: *"First, both the B-A age and B-P age are considerably older at the LGM (~20 ka) than the preindustrial period"*

"Considerably" is rather vague and should be quantified or be removed.

Corrected

L82: *"This new approach of BwP age is considerably similar to the true ocean ventilation age globally"*

(1) The new approach is still unpublished, and Fig. 1 only shows that its results (but not that the various methods) are similar.

This is precisely what figure 1 shows, that BwP age derived using the new approach is very similar to the ideal age. The new approach itself (including the evaluation using observations) is beyond the scope of this work and the main conclusions of this work do not depend on the manuscript in review (Du et al).

(2) "Considerably" is rather vague and should be quantified or be removed.
Corrected.

(3) The "true" ocean ventilation age is the "ideal" model age (IAGE). Therefore, it would be more appropriate to rephrase, e.g., "These new BwP ages are similar to the ideal ocean ventilation tracer ages ..."

Revised as suggested. This is clarified in the revised manuscript: "This new approach of BwP age is similar to the ideal age (IAGE) globally" in line 82-83.

L194: *"dramatic bell shape"*
"Dramatic" should be removed.
Corrected.

L195-196: *"A similar bell shape pattern is also seen for the Pacific and Atlantic mean"*
There is definitely no bell shape in the Atlantic but a clear age drop near 14 ka BP.
This is now clarified in the revised manuscript: "A similar bell shape pattern is also seen for the Pacific mean (Fig. 1b)" in line 195-196.

L209-210: *" (…) the decrease in global AABW transport aligns with the increase in global mean IAGE during the same period (14–13 ka)."* According to Fig. 1d, AABW abruptly increases between 14-13 ka BP, that is, before the age peak at 12.9 ka BP. What is the explanation for this lag?
This is now explained in the revised manuscript: "As shown in figure 1, starting from 19 ka,

the GMOC and AABW both fall rapidly and then shows a sharp recovery. When the AABW transport reaches its minimum strength at approximately 14 ka, the global mean IAGE increases towards its maximum at 12.9 ka. The lag between the AABW transport minimum at 14 ka and IAGE peak at 12.9 ka is likely because the "memory" of ocean typically last thousands of years. That is, the response time scale for the slow evolution of circulation associated with the AABW and abyssal flows can be over a thousand years. As such, the decrease in global AABW transport aligns with the increase in global mean IAGE during the same period" in line 207-212.

L274-277: *"The mean IAGE for AABW water mass increases from 836 years at the LGM to 1813 years at 14 ka due to the weakening of AABW transport, while the age of NADW increases by up to 1500 years during the period of 12–11 ka due to more NADW sinking into deeper depths in the Arctic (Fig. 2j-l), resulting in a relatively older water age for NADW and the lag of 2000 years between the maxima of Dye_NA and Dye_S (Fig. 3a-b)"*.
(1) It would be more obvious to show a time series of NADW strength which could be added to Fig 1d.
Actually, the time series of NADW is shown as GMOC in Figure 1d, which is diagnosed as the maximum in the GMOC streamfunction below 600 m from 33° S-60° N. Therefore, the main feature of the GMOC follows the upper clockwise cell mostly confined to the Atlantic sector.

(2) How is it possible that enhanced NADW production, i.e. enhanced ocean ventilation can lead to water mass ageing?
In order to avoid confusion, this sentence is now revised in Lines 280-282: "The increased water age of NADW is attributed to a greater fraction of NADW sinking into deeper Arctic depths (Fig. 2j-l), resulting in an increased water age for NADW as distance from the formation regions increases and the lag of 2000 years between the maxima of Dye_NA and Dye_S".

(3) According to Fig 3b the volume of North Atlantic waters decreases between 14-12 ka. This seems to contradict more NADW sinking.
We believe the revised texts (see our response above) improves the clarity. Figure 3b shows the total volume of NADW below 1 km in the interior ocean, and it does not necessarily mean enhanced NADW production. We see how our original language could be confusing, which is now revised in Lines 280-282.

(4) I am not sure of how to interpret Fig. R3 in the author's reply to my first review. What are the units? It appears that DYE_NA at 2 km declines before the maximum of IAGE in Fig. 3a in the paper.
The unit is percentage, similar to Gu et al. (2020). The intention of figure R3 is to show an example of more percentage of NADW water mass sinking into greater depths (up to 3.9 km) in the Arctic since 14.5 ka. As NADW water mass sinks into deeper depth, the water age of NADW generally increases with distance from the formation regions. As such, the IAGE of NADW increases to its maximum during 12 – 11 ka in figure 3a. This is now revised in Lines 280-282.

L333-334 (and L315 in the initial submission): *"The calculated southward DWBC and northward AABW DWBC are validated by the model MOCs, which are defined the same as*

*in the C-iTRACE"*. You cannot validate model results with model results obtained with the same model and in the same simulation. If the results were not consistent something would be wrong.

This is clarified in the revised manuscript "The calculated southward DWBC and northward AABW DWBC are consistent with the model MOCs" in line 338-339. In this work, the model MOCs are calculated based on complete velocity analyses from top to bottom at numerous latitudes, while the calculated northward DWBC and northward AABW across different basins in this work are only calculated using meridional velocity at 30° S. In other words, MOCs are "ground truth" in this context, and therefore we believe the comparison is necessary to access the accuracy of the calculations.

Reference:

Gu, S., Liu, Z., Oppo, D. W., Lynch-Stieglitz, J., Jahn, A., Zhang, J., and Wu, L.: Assessing the potential capability of reconstructing glacial Atlantic water masses and AMOC using multiple proxies in CESM, Earth Planet. Sci. Lett., 541, 116294, https://doi.org/10.1016/j.epsl.2020.116294, 2020.